# Changes in Fatty Acid Dietary Profile Affect the Brain–Gut Axis Functions of Healthy Young Adult Rats in a Sex-Dependent Manner

**DOI:** 10.3390/nu13061864

**Published:** 2021-05-30

**Authors:** Damian Jacenik, Ana Bagüés, Laura López-Gómez, Yolanda López-Tofiño, Amaia Iriondo-DeHond, Cristina Serra, Laura Banovcanová, Carlos Gálvez-Robleño, Jakub Fichna, Maria Dolores del Castillo, José Antonio Uranga, Raquel Abalo

**Affiliations:** 1Department of Cytobiochemistry, Faculty of Biology and Environmental Protection, University of Lodz, 90-236 Lodz, Poland; damian.jacenik@biol.uni.lodz.pl; 2Department of Basic Health Sciences, Faculty of Health Sciences, University Rey Juan Carlos (URJC), 28922 Alcorcón, Spain; ana.bagues@urjc.es (A.B.); laura.lopez.gomez@urjc.es (L.L.-G.); yolanda.lopez@urjc.es (Y.L.-T.); crisserra80@gmail.com (C.S.); banovcanova.laura@gmail.com (L.B.); carlos.galvezr@urjc.es (C.G.-R.); jose.uranga@urjc.es (J.A.U.); 3High Performance Research Group in Experimental Pharmacology (PHARMAKOM-URJC), URJC, 28922 Alcorcón, Spain; 4Associated I+D+i Unit to the Institute of Medicinal Chemistry (IQM), Scientific Research Superior Council (CSIC), 28006 Madrid, Spain; 5High Performance Research Group in Physiopathology and Pharmacology of the Digestive System (NeuGut-URJC), URJC, 28922 Alcorcón, Spain; 6Food Bioscience Group, Department of Bioactivity and Food Analysis, Instituto de Investigación en Ciencias de la Alimentación (CIAL) (CSIC-UAM), Calle Nicolás Cabrera, 9, 28049 Madrid, Spain; amaia.iriondo@csic.es (A.I.-D.); mdolores.delcastillo@csic.es (M.D.d.C.); 7Department of Biochemistry, Faculty of Medicine, Medical University of Lodz, 92-215 Lodz, Poland; jakub.fichna@umed.lodz.pl; 8Working Group of Basic Sciences in Pain and Analgesia of the Spanish Pain Society (Grupo de Trabajo de Ciencias Básicas en Dolor y Analgesia de la Sociedad Española del Dolor), 28046 Madrid, Spain

**Keywords:** behavior, brain–gut axis, coconut oil, diet, fatty acids, gastrointestinal motility, primrose oil, sex, soy oil, visceral pain

## Abstract

Dietary modifications, including those affecting dietary fat and its fatty acid (FA) composition, may be involved in the development of brain–gut axis disorders, with different manifestations in males and females. Our aim was to evaluate the impact of three purified diets with different FA composition on the brain–gut axis in rats of both sexes. Male and female Wistar rats fed a cereal-based standard diet from weaning were used. At young adult age (2–3 months old), animals were divided into three groups and treated each with a different refined diet for 6 weeks: a control group fed on AIN-93G diet containing 7% soy oil (SOY), and two groups fed on AIN-93G modified diets with 3.5% soy oil replaced by 3.5% coconut oil (COCO) or 3.5% evening primrose oil (EP). Different brain–gut axis parameters were evaluated during 4–6 weeks of dietary intervention. Compared with SOY diet (14% saturated FAs, and 58% polyunsaturated FAs), COCO diet (52.2% saturated FAs and 30% polyunsaturated FAs) produced no changes in brain functions and minor gastrointestinal modifications, whereas EP diet (11.1% saturated FAs and 70.56% polyunsaturated FAs) tended to decrease self-care behavior and colonic propulsion in males, and significantly increased exploratory behavior, accelerated gastrointestinal transit, and decreased cecum and fecal pellet density in females. Changes in FA composition, particularly an increase in ω-6 polyunsaturated FAs, seem to facilitate the development of brain–gut axis alterations in a sex-dependent manner, with a relatively higher risk in females.

## 1. Introduction

Epidemiological and clinical studies demonstrate that diet affects many aspects associated not only with physiological but also pathophysiological conditions. Amongst the macronutrients, fat has been the focus of a vast number of studies. Thus, it is well known that high-fat diets (HFDs) have deleterious effects on the cardiovascular and endocrine systems [1,2], although their impact on other systems, such as the gastrointestinal (GI) tract (motility, sensitivity) and the central nervous system (behavior), i.e., the brain–gut axis, has attracted less attention. Furthermore, the impact of changing particular dietary fat components has been less studied.

Fatty acids (FAs) are the main components of triglycerides and are classified as saturated fatty acids (SFAs), monounsaturated fatty acids (MUFAs), and polyunsaturated fatty acids (PUFAs) according to the presence and number of double bonds in their molecular structure. FAs are the second most important source of energy for the human organism and a crucial component of cellular membranes. FAs seem to be a major factor influencing the development of cardiovascular disease, rheumatoid arthritis, neurodegenerative diseases, and immune-related pathologies such as metabolic diseases, chronic obstructive pulmonary disease, and some cancer types [3,4,5,6,7,8,9]. However, so far, only limited efforts have been made to learn how the composition of FAs in the diet affects behavior and GI tract function, which are key to understanding the etiopathology of functional GI disorders (FGIDs), now considered as brain–gut axis disorders [10].

One of these FGIDs, or brain–gut axis disorders, is irritable bowel syndrome (IBS). The main symptoms of IBS are visceral hypersensitivity, altered intestinal motility and mucosa permeability, as well as immune dysfunction in the colon, and this functional disorder is generally associated with emotional alterations like depression or anxiety. Prevalence of IBS seems to be higher in females [11], and it often occurs with comorbidities like headaches, fibromyalgia, or other chronic pain syndromes [11]. Interestingly, clinical trials have shown that ω-3 PUFAs can reduce depression [12], whilst in rodents their deficiency can increase depression and anxiety behaviors [13,14], and HFDs rich in SFAs induce anxiety and anhedonia behaviors in rats [15]. Furthermore, obesity has been associated with other comorbid pathologies such as headaches, fibromyalgia, or arthritis [16,17]. In animal models, HFDs worsened inflammation-induced pain [18], although less is known about the impact of particular FAs on tactile or skeletal muscle sensitivity. Importantly, IBS and other FGIDs do not involve gross pathological findings in the gut wall, compared with inflammatory diseases of the GI tract [19].

Within the GI tract, a positive association between fat consumption, mainly trans-unsaturated and ω-6 FAs, has been shown for ulcerative colitis [20,21]. Basic research has demonstrated that a diet enriched in SFAs alters intestinal permeability and microbiota profile and increases the expression of inflammatory markers in the colon [22], whilst diets rich in ω-6 or ω-3 PUFAs do not seem to affect colonic permeability nor mesenteric fat inflammation [23]. However, other studies have shown that diets rich in unsaturated FAs can also have deleterious effects, for example, ω-6 PUFAs increase pro-inflammatory bacteria and colitis susceptibility in mice [24]. In addition, mice fed a HFD mainly from soybean oil presented a more affected crypt length in the proximal colon than mice fed a HFD rich in coconut oil [25]. 

As pointed out, most of these studies have been performed using HFDs (over 40% energy from fat in the rodent diet, [26,27]), but normocaloric or slightly hypercaloric diets (around 8–16% energy from fat in the rodent diet) might be more translatable for FGIDs from a clinical point of view. One study exposed young adult male and female rats (7 weeks old) to AIN-93G diet, a purified diet with 16% energy from fat, recommended by the American Institute of Nutrition for rapid growth, pregnancy, and lactation [28], and considered it as an appropriate diet for use in rat safety evaluation studies, although functional parameters affecting the brain–gut axis were not analyzed [29]. Furthermore, little is known about how the different types of dietary FAs can alter GI motility and visceral sensitivity and consequently contribute to the development of IBS or other FGIDs. In this line, Mosińska et al. used purified diets with 16% energy from fat and different FA composition (AIN-93G and two additional diets modified from it, one enriched in SFAs and the other one enriched in PUFAs) in male rats [30] and mice [31], but females were not studied. Furthermore, the effects of these diets on behavioral parameters relevant to the occurrence of IBS and comorbid chronic pain syndromes are unknown.

Thus, we aimed to evaluate the changes produced by these purified diets, with 16% energy from fat and different FA composition, on the brain–gut axis functionality (i.e., emotional behavior parameters as well as GI tract motility and sensitivity) in both male and female rats.

## 2. Materials and Methods

### 2.1. Ethics Statement

The experimental protocol was approved by the Ethic Committee at the University Rey Juan Carlos (URJC) and Comunidad Autónoma de Madrid (PROEX 063/18, PROEX 023/19) and complied with the European Community Council Directive of September 22, 2010 (2010/63/EU) and the Spanish regulations (Law 32/2007, RD 53/2013 and order ECC/566/2015). Every effort was made to minimize animal pain and discomfort as well as to reduce the number of animals used.

### 2.2. Animals

Young adult (2–3 months old) female (*n* = 36, 166–231 g body weight) and male (*n* = 44, 275–305 g body weight) Wistar rats were obtained from the Veterinary Unit (URJC, Spain). Animals were housed in standard transparent cages (60 × 40 × 20 cm; 3–4 animals per cage) at a constant temperature (20 °C) and relative humidity (60%) and maintained under a 12 h light/dark cycle (lights turned on at 8 a.m.) with free access to chow pellets (SAFE D40 diet (www.safe-diets.com, accessed on 27 May 2021)) and sterile tap water.

### 2.3. Diets

Rats were randomly allocated to three experimental groups. The first group of animals received AIN-93G diet, a validated purified diet [30] containing 7% of soybean oil as the only source of fat (SOY; ZooLab, Sędziszów, Poland), which was used as control diet in this study. The second and third groups of animals were fed on AIN-93G diets containing 3.5% of soybean oil and supplemented with 3.5% of coconut oil (COCO; ZooLab) or 3.5% of evening primrose oil (EP; ZooLab), respectively. 

All diets were formulated to meet nutritional requirements of young adult rats. The ingredients of each diet and the type of FAs are summarized in Table 1A,B, respectively (the composition in fatty acids of soybean oil, coconut oil, and evening primrose oil can be found in Appendix A).

### 2.4. Experimental Protocol

All rats used in the study were born, weaned, and maintained in our own animal house facilities. Male and female rats were obtained at a young adult, sexually mature age (2–3 months old) and acclimatized for one week in our experimental laboratory, fed on the same cereal-based (non-purified), standard laboratory diet that had been their normal diet from weaning. Afterwards, the rats were randomly allocated to the three different purified diets (Figure 1). 

Exposure to the diets started on week 1 of the study and lasted for 6 weeks. Different parameters were evaluated (Figure 1). Body weight and food and water intakes were regularly monitored during the whole study. Brain functions were evaluated by means of behavioral tests during week 4 of diet exposure. Tests for GI motor function and colonic sensitivity were performed during weeks 5 and 6, respectively. To minimize circadian influence, experiments were performed between 8:00 a.m. and 12:00 a.m.; X-ray sessions started between 8.00 a.m. and 10.00 a.m. All experimental procedures were performed and analyzed blindly with respect to diet type.

### 2.5. General Health Parameters

#### 2.5.1. Body Weight, Food, and Water Intake

Body weight was recorded three days a week during the six weeks of feeding on the experimental diets (Figure 1) and before sacrifice at the end of the study. Food and water intakes were measured manually every other day throughout the six weeks of feeding. Remnants of chow pellets were not reused. Food and water intakes are represented as the combined data from all animals belonging to each cage in each experimental group and is shown as the average daily intake per rat. Body weight gain is presented normalized to food intake as the ratio weight gain/food intake.

#### 2.5.2. Plasma Levels of Glucose, Cholesterol, and Triglycerides

Plasma was collected, without prior fasting, from 6 animals per group (not previously used for visceral sensitivity experiments), at sacrifice. Determinations of plasma levels of cholesterol, triglycerides, and glucose were performed spectrophotometrically using commercial kits (Spinreact, Girona, Spain). Results were normalized to the average food intake of each group of animals and are expressed as mg/dL/g of food intake. 

### 2.6. Brain Functions

#### 2.6.1. Splash Test

At the beginning of the fourth week (Figure 1), the splash test was performed as described by Csongová et al. [32]. The splash test was used to estimate the grooming behavior, described as cleaning of the fur of the animal by licking and/or scratching after spraying a 10% sucrose solution onto the rat’s dorsal coat. Grooming events were recorded including nose and face grooming (e.g., strokes along the snout), head washing (e.g., semicircular movements over the top of the head and behind the ears) and body grooming (e.g., body fur licking). The latency and the duration of grooming were recorded during 5 min. after spraying the sucrose solution.

#### 2.6.2. Hole-Board Test

The hole-board test was applied at least 60 min. after the splash test (Figure 1), to estimate exploratory behavior and neophilia as well as neophobia. Neophilia is defined as the attraction that animals display towards a novel object or place, while neophobia is the aversion that animals show towards an approaching and unknown object or place. The hole-board test box consists of a box, whose floor (60 × 60 cm), which is slightly raised, has a total of four holes placed in two lines, and all of them are equidistant to each other and to the walls (40 cm height). In addition, the floor is divided into 36 squares (10 × 10 cm each), distributed 6 squares across and 6 squares down. At the beginning of the session, each rat was removed from its home cage and placed in the same corner of the hole box; the rat behavior was recorded for 10 min. (during this time the researcher was not in the room). The following behaviors were recorded off-line: number of external and internal squares that the rat crossed (spontaneous movement); amount of time that the rat placed its head into one of the holes, to a minimum depth such that the ears were at the same level as the floor of the box (head-dip); amount of time the animal remained stationary and raised its fore limbs off the ground, standing vertically on its hind limbs (rearing); amount of time that the animal was grooming; amount of time that the rat was still; number of feces expelled during the test [33].

#### 2.6.3. Somatic Sensitivity

Different tests were applied at the end of week 4, accordingly with Figure 1, to evaluate changes in somatic sensitivity, as described below.

##### Von Frey Test

The Von Frey test was performed to estimate mechanical sensitivity to non-noxious mechanical stimuli. Rats were placed separately on an elevated iron mesh covered by a transparent cage (10 × 20 × 15 cm) and were allowed to adapt to the environment for at least 15 min. Habituation to this environment was also performed two days before assessment. Mechanical sensitivity was assessed by measuring the withdrawal threshold to a series of calibrated von Frey filaments (6–26 g; Ugo Basile). The nociceptive test was done by inserting the von Frey filament through the mesh floor and applying it to the plantar surface of each hind paw. Each stimulus was applied five times per paw with an interstimulus interval of approximately 30 s. A positive result was considered when at least three of the five trials induced a withdrawal response with the same filament. The lowest filament that induced a positive result was considered as the mechanical threshold, and the results obtained in both paws were averaged and used for data analysis [34].

##### Plantar Test

Plantar test was conducted to determine the sensitivity threshold to heat noxious stimuli [35]. Responses to thermal stimuli were evaluated immediately after the von Frey test using Hargreaves 37,370 apparatus (Ugo Basile). Rats were placed separately on an elevated glass floor in a transparent cage (86 × 40 × 35 cm) and were allowed to adapt to the environment for at least 15 min. the day of experimentation and the two previous days. The withdrawal latency from a focused beam of radiant heat applied to the mid plantar surface of the hind paws was recorded. The intensity of the light was adjusted at the beginning of the experiment so that the control average baseline latencies were about eight seconds; a cut-off latency of 25 s. was imposed. The withdrawal latency of each paw was measured during three trials separated by two-minute intervals, and the mean of three readings was used for data analysis.

##### Pressure Administration Device Test

To determine the sensitivity threshold of the skeletal muscle to noxious mechanical stimulation, the pressure administration (PAM) device was used in the gastrocnemius muscle, as previously described [36]. The PAM device consists of a force transducer mounted on a unit fitted to the operator’s thumb with an eight mm diameter circular contact. The rat was gently held by the researcher, wrapped with a cloth, leaving the hindlimb accessible; the rat was accustomed to being held for two days before experimentation began. The force applied to the gastrocnemius muscle was increased at a constant rate of 50 g per s. until the animal withdrew its hindlimb or vocalized or a cut-off pressure of 400 g/force (g/f) was reached. The peak g/f applied immediately prior to hindlimb withdrawal or vocalizing was recorded, and the mean of three consecutive trials was considered the nociceptive mechanical threshold.

#### 2.6.4. Locomotor Activity Analysis

Locomotor activity was evaluated using individual photocell activity chambers after somatic sensitivity evaluation, at the end of the fourth week (Figure 1) as described previously [37]. Each rat was placed separately in the recording chamber (Cibertec S.A, Madrid, Spain; 55 × 40 cm; three cm spacing between beams) where the number of interruptions of photocell beams was recorded over a 30 min. period. The mean number of crossings of the photocell beams was used for comparison.

### 2.7. Gastrointestinal Functions

#### 2.7.1. Gastrointestinal Motility Analysis

GI motor function was studied using radiographic methods as previously described [38]. Five weeks after initiation of diet intervention (Figure 1), 2.5 mL of barium sulfate (Barigraph^®^AD, Juste SAQF; 2 g/mL, temp. = 22 °C) was administrated per os through curved gavage needles (14 gauge, 2.9 inch length, and 4 mm ball diameter). Radiographs of the GI tract were obtained at different time points, i.e., immediately (0) and 1, 2, 4, 6, 8, and 24 h after administration of contrast medium. Radiographs were recorded on Carestream Dental T-MAT G/RA film (15 × 30 cm) housed in a cassette provided with regular intensifying screen using a CS2100 digital X-ray apparatus (Carestream Dental; 60 kV, 7 mA). Exposure time was adjusted to 20 ms, and focus distance was manually fixed at 50 ± 1 cm. Rats did not receive anesthesia and were immobilized in the prone position inside adjustable, hand-made, transparent plastic tubes. To reduce stress, rats were habituated to stay within the plastic device for the recording period and were released immediately after each X-ray shot (immobilization lasted for up to two minutes). Habituation to the recording chamber prior to commencement of the study did not significantly alter GI motility [38]. Alterations in GI tract motility were semiquantitatively determined from the X-ray images by assigning a compounded value to each region of the GI tract considering the parameters described in Table A2 (Appendix B). In addition, X-rays were scanned, and the size as well as density of the stomach, small intestine, cecum, and fecal pellets were analyzed using Image J 1.38 for Windows (National Institute of Health) [39].

#### 2.7.2. Colonic Bead Expulsion Test

Two or more days after the X-ray analysis (Figure 1), colonic bead expulsion test was performed as previously described [40]. Briefly, a glass bead (diameter: 8 mm) with a fire polished end, pre-warmed to 37 °C, was inserted 3 cm into the distal colon. To facilitate insertion and avoid tissue damage, beads were covered with Vaseline. Then, the animals were placed into individual transparent cages, and the time to bead expulsion was measured with a cut-off limit of 4 h (14,400 s) [41]. Thus, animals were monitored for a maximum of four hours, unless the bead expulsion occurred sooner. Sedation with Sedator^®^ (medetomidine hydrochloride, an α_2_ adrenergic agonist; 1 mL/kg, 1 mg/mL; ip) was used to insert the bead into the colon, and Revertor^®^ (atipamezole hydrochloride, an α_2_ adrenergic antagonist; 0.66 mL/kg, 5 mg/mL; ip) was used to revert the sedation [42].

#### 2.7.3. Colonic Sensitivity Analysis

On the sixth week of diet intervention (Figure 1), approximately half of the rats used in each experimental group (6 females and 8 males per group) were used for this study. After sedation with Sedator^®^ (1 mL/kg, 1 mg/mL; ip), a 10 cm longitudinal line over the linea alba of the rat abdomen and transverse lines every 2 cm were drawn [43]. Feces were gently removed from the rectum, and a 5 cm long latex balloon lubricated with Vaseline was inserted into the colon so that its tip was located 7 cm inside the colorectum. Sedation was reverted with Revertor^®^ (0.66 mL/kg, 5 mg/mL; ip) and rat behavior recorded for 35 min using a video camera located 30 cm below the cage floor. After 5 min, the pressure of the intracolonic balloon was increased, using a sphygmomanometer and phasic stimulation was applied as previously described [44]. With this type of stimulation, the same 20 s long stimulus was repeated three times within each 5 min-period, with a 1 min stimulus-free period between individual stimuli. Pressure was increased from 0 to 80 mm Hg, in steps of 20 mm Hg every 5 min., to finally return to 0 mm Hg. The videos were exported as a series of frames (1 s) using Quick Time Player Pro for Windows (v.7.7.4; Apple Inc., Cupertino, CA, USA). Using these frames, the average number and duration of contractions as well as the percentage of time that the rat spent contracting the abdomen were evaluated for each pressure stimulus.

### 2.8. Macroscopic Analysis

After six weeks of feeding, 6 animals per group (not previously used for visceral sensitivity experiments) were weighed and sacrificed (Figure 1) by cervical dislocation and exsanguination, and white fat pads, stomach, small intestine, cecum, and colorectum were separated and weighed. The small intestine and colorectum were weighed also after content removal. Small intestinal content was “milked” as previously described by Robert et al. [45] and weighed. The colon was gently flushed with saline. The size of the stomach and cecum, as well as the length of small intestine and colon, were measured before contents removal. Fat pads, ileum, and distal colon samples were collected for further analysis.

### 2.9. Histological Analysis

Tissues for histology were obtained from the same animals that underwent macroscopic analysis (6 per group, not used for visceral sensitivity experiments). White fat pads, ileum, and colon tissues were fixed in buffered 10% formalin and embedded in paraffin. Subsequently, 5 µm sections were stained with hematoxylin and eosin (HE) or toluidine blue and studied under a Zeiss Axioskop 2 microscope equipped with the image analysis software package Axio Vision 4.6. The mean area of adipocytes was measured with the open automated software Adiposoft for Fiji (Image J) in three sections per animal under a 20× objective [46]. The analysis of ileal and colonic damage was made in ten random fields per section measured under a 40× objective with a total of three independent sections of the ileum/colon tissue per animal. Histological damage score for the ileum was assessed by criteria adapted from Galeazzi et al. [47] based on the presence (score = 1) or absence (score = 0) of crypt abscess formation and goblet cell depletion, loss of mucosal architecture (0 = normal, 1 = moderate, 3 = extensive), the extent of inflammatory cell infiltrate (0 = normal, 1 = moderate, 3 = transmural), and muscular layer thickness (0 = normal, 1 = reduced). The histological damage to the colon was evaluated according to Saccani et al. [48], as the sum of the following parameters: epithelial damage (0 = normal, 1 = moderate, 3 = extensive), infiltration of inflammatory cells (0 = absence, 1 = normal, 3 = extensive, 4 = extensive involving submucosa), separation of muscle layer and muscularis mucosae (0 = normal, 1 = moderate, 2= extensive), and goblet cell depletion (0 = absence, 1 = normal, 3 = extensive, 4 = very extensive). Thus, maximum damage to the ileum and colon was represented by 9 and 13 points, respectively. The width of circular and longitudinal muscle layers was measured [49]. The number of mast cells stained with toluidine blue was counted under a 40x objective in ten fields per animal all along the area between the epithelium and external muscular layer. A quantitative analysis of mast cells was made in three independent sections of the colon tissue per animal [30].

### 2.10. Statistical Analysis

The sample size needed for the study was calculated with the program G*power, taking into consideration our previous studies. Thus, with an α error 0.05 and power = 0.8, and considering an effect size of 0.25 as relevant, the result suggested that 6 animals were needed in each experimental group. Therefore, we considered 6 as the minimum number of animals per experimental group (allocated to the different tests) necessary to obtain reliable results in our study.

Statistical analysis was performed using GraphPad Prism 7.0 (GraphPad Software Inc., San Diego, CA, USA). Data were tested for normal distribution using the Shapiro–Wilk normality test. Data are presented as the mean ± standard deviation (S.D.) when normally distributed, or median ± interquartile range (IQR) when not normally distributed. To compare the normally distributed data, one or two-way ANOVA were used, followed by Bonferroni’s multiple comparison post hoc test; in the case of not normally distributed data, Kruskal–Wallis test followed by Dunn´s multiple comparison test was performed. Values of *p* < 0.05 were considered statistically significant. The effects of diet intervention were evaluated in males and females separately (no inter-sex comparison was made).

## 3. Results

### 3.1. General Health Parameters

The initial weight of male and female rat groups allocated to each diet is presented in Appendix B (Table A1). The statistical analysis did not show any significant difference among the female groups, whereas the group of animals that was later exposed to EP diet showed a significantly higher average body weight (*p* < 0.01) than those later fed on SOY and COCO diets. Males exposed to SOY and COCO diets did not show any significant difference in their initial average body weight. The body weight of the animals under study was within the range expected for young adult (2–3 months old) Wistar male and female rats [50].

There were no statistically significant differences across diets in food intake, neither in males nor females, although there tended to be an increased intake in males treated with EP diet (Appendix A). Regarding water intake, both COCO and EP diets tended to increase this value in both males and females, but the differences with SOY did not reach statistical significance (Appendix A). To avoid the influence of the differences in food (energy) intake, body weight gain was presented normalized to food intake (Appendix A). As can be seen in Appendix A, no differences were found between the different groups of females nor between males treated with COCO when compared to SOY, although EP male rats did present an increase in weight gain when compared to the group fed on SOY (*p* < 0.05).

Diet intervention with COCO or EP did not significantly affect the plasma level of glucose nor cholesterol in either male or female rats compared with the animals exposed to SOY diet (Appendix A). Regarding plasma triglycerides, the only difference with SOY-fed animals was found in males under EP diet, which presented statistically significant lower values (Appendix A), whilst this difference was not found in females.

The weight of white adipose tissue normalized to body weight gain was not significantly modified by exposure to COCO or EP diets compared with SOY diet in either sex (Appendix A). Finally, based on HE staining of sections obtained from adipose tissue, neither sex nor diet altered in a significant manner the adipocyte area (Appendix A). Representative images are shown in Appendix A.

### 3.2. Brain Functions

In the splash test, two different parameters were evaluated: the time that the animals took to start grooming (latency) (Figure 2A) and the time they spent grooming (Figure 2B). Altogether, compared with SOY diet, COCO and EP diets did not significantly modify the latency nor duration of grooming in the splash test in either sex; however, there tended to be a decreased grooming behavior in males fed on EP diet when compared to those in the SOY diet group (*p* = 0.06).

When analyzing the differences in the hole-board test (Figure 3), exposure to COCO or EP diets did not produce any statistically significant variations compared with SOY in males. On the contrary, in females, EP diet significantly increased the number of crossed squares (*p* < 0.05) (Figure 3A) and the time that animals stood on their hindlimbs (*p* < 0.01) when compared with SOY diet (Figure 3E).

Compared with SOY diet, COCO and EP diets did not modify any of the somatic sensitivity thresholds (Figure 4) or locomotor activity (Figure 5) in either sex.

### 3.3. Gastrointestinal Functions

#### 3.3.1. Radiographic Analysis of Gastrointestinal Motor Function

##### Semiquantitative Score of Gastrointestinal Motility

Figure 6 shows the semiquantitative analysis of X-ray images for male and female rats exposed to SOY, COCO, and EP diets.

Compared with SOY, COCO and EP diets did not modify in a substantial manner GI motility in males, except for the fact that the emptying of the stomach and the small intestine was slightly slower in COCO and EP animals, respectively. In addition, arrival of barium to the cecum was slower in both COCO and EP animals, and was also slightly retarded in its arrival to the colorectum in COCO rats.

In females, compared with SOY diet, the emptying of barium from the stomach was slightly faster in EP animals, whereas in the small intestine it was slightly retarded in the COCO group. When compared to the SOY group, the arrival of barium to the cecum was slightly slower in COCO, but its arrival was slightly accelerated to the colorectum in the EP group.

##### Morphometric and Densitometric Analysis of Stomach and Cecum

As expected, the curves representing the changes in size and density of stomach and cecum along time (Figure 7) were similar to those obtained in the semiquantitative analysis (Figure 6). However, there was a slight but significant decrease in the maximum size of the stomach of male rats in the COCO group (Figure 7A) and an increase in the maximum density of the cecum content in the COCO and EP female groups (Figure 7D’), when compared to their respective SOY groups.

##### Number, Size, and Density of Fecal Pellets

A progressive increase in the number of fecal pellets from T4 until the end of the experiment was observed in males and females under SOY, COCO, and EP diets (Figure 8A,A’).

Compared with SOY, COCO and EP diets did not significantly modify the parameters of the fecal pellets in males (Figure 8A–D).

On the contrary, in EP females, there was an increase in the number of fecal pellets from T2 until T8 and a decrease in T24 (Figure 8A’), and fecal pellet size was smaller in a significant manner at T8 (Figure 8B’), with no differences in fecal diameter (Figure 8C’). Fecal pellet density was increased in EP and COCO groups compared with SOY group at all of the time points in both sexes, although this difference was statistically significant only in females (Figure 8D,D’).

#### 3.3.2. Colonic Propulsion

The latency to expel a bead inserted 3 cm in the colon was not significantly modified by the diet in either sex (Figure 9).

#### 3.3.3. Colonic Sensitivity

There was a progressive increase in the number of contractions in response to an increasing mechanical stimulation in both female and male rats (Figure 10).

Compared with the SOY diet group of the corresponding sex, COCO and EP diets produced a similar number of contractions (Figure 10A,A’), although the contraction duration was reduced in the COCO male group at 60 mmHg (Figure 10B). The percentage of time spent in contraction was similar for the three diets, without statistically significant differences in either sex (Figure 10C,C’).

### 3.4. Macroscopic Features of the Gastrointestinal Organs

For an easier comparison, the weights of the organs were normalized to body weight (Appendix A). Compared with SOY diet, no significant modifications were induced by COCO and EP diets on organ weights or sizes. In contrast, in females under EP diet, a significant increase in normalized stomach weight and a reduction in cecum size and colorectum length compared with the corresponding SOY group was found (Appendix A).

### 3.5. Microscopic Features of Gut Wall

Irrespective of the sex of the animals, neither COCO nor EP diets significantly altered the histological damage score in either ileum or colon when compared to SOY diet, which was low in all animals (Table 2).

The other parameters measured in the gut wall of males and females (colonic longitudinal and circular small muscle layer thickness and density of colonic mast cells) were not significantly affected by COCO and EP diets either (Table 2). Representative images for gut wall microscopic studies are shown in Appendix A.

## 4. Discussion

In the present study, we examined the effect of diet intervention using modified diets, characterized by different composition of FAs, on GI motility, behavioral (emotional) parameters, and visceral and somatic pain perception, all relevant to evaluate the occurrence of brain–gut disorders and comorbid chronic pain syndromes, in both male and female healthy young adult rats.

Specifically, we used three different refined diets, i.e., SOY, COCO, and EP diets, containing the same amount of fat (7%) that accounted for 16% energy (thus, these diets cannot be considered HFD, which contain around or above 40% energy from fat, [26,27]), but different FA composition (Table 1). Soybean oil, the only source of fat in SOY diet, contains 51% of linoleic acid (PUFA, C18:2) and about 7% of α-linoleic acid (PUFA, C18:3), as well as monounsaturated FAs (~23% of FA content). In COCO diet, 50% of soybean oil was substituted by coconut oil; thus, COCO diet had 52% SFAs, where lauric acid (SFA, C12:0) represents 23.4% of its fat content. Finally, in EP diet, 50% of soybean oil was substituted by evening primrose oil; in this diet, the main FA was linoleic acid (62.44%). As summarized in Table 3, compared with SOY diet, these changes in the dietary FA content produced relatively mild modifications in the parameters evaluated, but, interestingly, these were clearly sex-dependent, and those associated with exposure to EP diet were more prominent than those associated with exposure to COCO diet.

AIN-93G diet was developed to use during stages of rapid growth, pregnancy, and lactation [28], and it has rarely been used in studies which evaluate its impact on young adult, sexually mature animals. A previous report exposed young (7 weeks old) male and female Sprague-Dawley rats to AIN-93G diet (SOY diet, used as control diet in the present study) and showed normal growth and development (comparable to standard, cereal-based chow) after 6 and 13 weeks of feeding with this diet, without toxicologically relevant findings, but with evidence of improved feed efficiency as well as hypercholesterolemia (only in males) and hypertriglyceridemia, probably associated with the fact that AIN-93G diet is a casein-based diet [29]. In agreement with this report, our results did not show any evidence of toxicity by exposure to any of the three purified diets tested here, SOY (AIN-93G), COCO, and EP. Furthermore, the change of diet at the adult age in animals did not affect their global health since the parameters used to evaluate growth (Appendix A) and metabolic functions (Appendix A) did not exceed the physiological values described for Wistar rats at this age and for both sexes [51,52].

### 4.1. Gastrointestinal Tract Motor Function Evaluation

In line with our previous report [30], in the radiographic study, the GI motility curves obtained in the semiquantitative analysis for males exposed to the three purified diets were very similar (except for specific small alterations, such as a slightly retarded emptying of barium from the small intestine in the EP diet group and slower entrance of barium into the cecum in the EP and COCO groups). Furthermore, as in that report [30], the three groups of males displayed a similar size of GI organs, evaluated either radiographically in vivo (Figure 6 and Figure 7) or macroscopically at sacrifice (Appendix A), and the maximum barium densities within the stomach, cecum, and fecal pellets were also similar when evaluated by X-ray (Figure 6, Figure 7 and Figure 8). In addition, in the colonic bead expulsion test, which specifically allows for colonic propulsion evaluation, no significant differences were seen across the three diets, as also observed in our previous report [30], although in both studies EP diet tended to increase the time of bead expulsion. Altogether, our results suggest that none of the diets used here induced major alterations in GI motor function and macroscopic structure of the GI tract in males. In addition, the gut wall architecture did not show significant differences among the three male groups, further supporting that these diets are not harmful to the intestine.

In our previous study, only males were used [30]. Here, we evaluated the effects of the purified diets also in females. Interestingly, compared with SOY-fed females, the semiquantitative radiographic analysis of GI motility produced also very similar results for the other two female groups, with some minor differences (see below). In addition, compared with SOY diet, both COCO and EP diets significantly increased the maximum density of barium in the cecum and fecal pellets in the radiographic study (suggesting that contents in these elements were more concentrated, and probably less hydrated).

In general, EP diet seemed to be able to produce more prominent changes in GI motor function in females. Thus, in COCO-fed females, small intestine emptying was slightly faster, but cecum filling was slightly slower in the radiographic study. In contrast, in EP-fed females, both gastric emptying and colorectum filling were clearly faster than in SOY-fed females. Furthermore, the number of fecal pellets within the colon increased during the study in the EP group, compared with the SOY group. At sacrifice, females fed on EP diet had heavier stomachs, but the cecum was smaller and the colorectum was shorter. The increased number of fecal pellets associated with EP diet in females could be due to a higher production of fecal pellets (which seems unlikely, since no significant difference in food intake was found among the three groups), but also to the retention of these due to constipation. Unfortunately, many females scored the maximum value in the bead expulsion test, preventing us to determine if colonic motor function was actually uncoupled in this specific assay. Macroscopic changes in the large intestine (smaller cecum and shorter colorectum) in the females under EP diet might have contributed to these functional results: a smaller cecum might underlie the increased density of its content in the radiographic study; the shorter colorectum might explain accumulation of fecal pellets and faster emptying of the colorectum that, compared with other diets, was on-going at 24 h after barium.

Finally, as in males, no important alterations were apparent in the microscopic analysis of gut (ileum, colon) wall samples, suggesting that the functional alterations found in this study do not have an inflammatory origin.

### 4.2. Behavioral Tests

#### 4.2.1. Emotional Alterations

In IBS and other FGIDs, GI dysmotility is accompanied by emotional alterations, including depression or anxiety.

In the present study, the splash test was used to detect sings of anhedonia, as a marker of depression. In this test, a longer latency and shorter duration of grooming are markers of anhedonia [33]. Compared with SOY diet, changes in FA content did not significantly modify behavior in this test, although males exposed to EP tended to display less grooming behavior. Using the hole-board test, we evaluated the exploratory behavior of rats towards a new environment, which is a useful approach to estimate anxiety-like behavior and stress in animals. In this test, no significant differences among the three diets were found for males or associated with COCO diet compared with SOY diet in females. In contrast, females under EP diet increased the number of crossed squares and time standing on hindlimbs, suggesting they were “more exploratory” than animals under SOY and COCO diets. However, Clouard et al. (2014) documented that a diet with a low content of linoleic acid or a diet with a low linoleic acid/linolenic acid ratio increased exploration and decreased anxiety-related behavior [53], which is in contrast with our study. Interestingly, Sakayori et al. (2016) found that intervention of dams with a diet containing ~10% SFAs, ~15% MUFAs, and ~75% PUFAs was responsible for enhanced offspring’s anxious behavior, which seems to be related to an imbalance in the ω-6/ω-3 ratio [54]. The above-mentioned observation suggests that not only ω-3 PUFAs are characterized by beneficial roles in the modulation of emotional behavior, but also ω-6 PUFAs and ω-6/ω-3 PUFAs ratio may be crucial in this. In any case, despite the ω-6/ω-3 ratio, EP diet is the one that contains the greatest proportions of PUFAs. Furthermore, it is important to note that those studies were performed in very young, developing animals, in contrast with our study performed in young adult rats. Thus, age of animals may be crucial to determine the impact of dietary FA content changes.

Altogether, compared with SOY diet, in young adult animals EP diet seems to produce sex-dependent modifications in behavior, with males showing less motivation for self-care and females somehow increasing exploratory behavior. Interestingly, previous studies have shown that the brain composition of FA is sexually dimorphic, with the male brain having more SFAs and depleted in ω-6 PUFA when compared to female brains in control situations, without differences in plasma FA concentrations [55]. Thus, our results may be due to different sex-dependent mechanisms involved in appropriately “using” FAs, which warrants further investigation.

#### 4.2.2. Somatic and Visceral Nociception

Studying somatic as well as visceral nociceptive thresholds was considered highly relevant in our research because several pain syndromes such as prostatic or chronic pelvic pain syndrome, fibromyalgia syndrome, chronic fatigue syndrome, and migraine often co-exist with brain–gut axis disorders [56].

We did not find any substantial changes in response to colonic distension associated with COCO and EP diets, compared with SOY diet, in either males or females, although both diets seemed to slightly reduce the duration of contractions in males and the frequency of contractions (particularly at low pressures) in females. Previous studies in male rodents have stablished that EP diet can increase the visceromotor response to colonic distension in mice [31] and reduce the response to the highest pressures in rats [30]. Differences regarding the time animals were fed on the different diets, or the methods used to perform the colonic distension, can contribute to the differences found across this and previous studies. On the other hand, although HFD has been shown to worsen pain induced by different pathologies such as postoperative pain [57] or arthrosis [58], we did not observe any differences between diets in the different somatic nociceptive tests. Thus, although differences can be found while using HFD or when previous pathologies are present, the relatively mild modifications in FAs composition of the current diets do not seem to be enough to alter the nociceptive thresholds (either visceral or somatic) with the experimental conditions used in this study.

### 4.3. Limitations

This study has several limitations. In our scheme of experiments, a diet intervention of only six weeks was evaluated. This relatively short-term duration of treatment may be associated with minor changes, and the impact of long-term exposure to diet with different compositions of FAs should be analyzed in future studies. On the other hand, our study employed relatively small groups of animals in some tests, which can generate random differences, and a higher number of animals in each group could make the statistical results more robust or, at least, clarify the tendencies. Finally, most parameters were evaluated just at the end of the study (some behavioral tests should not be performed more than once, in order to avoid bias due to memory formation [59]), and only body weight was recorded at baseline. However, body weight is an essential parameter in animal research, generally considered valid to confirm (or not) the healthy initial conditions of the experimental cohorts. Despite these limitations, the above-presented and discussed results suggest that composition of FAs seems to be crucial for both brain (at least, emotional behavior) and gut functions (at least GI motor function and colonic sensitivity).

## 5. Conclusions

Taken together, our study shows that short-term intervention in healthy, sexually mature animals using non-high-fat diets with different FA compositions is able to generate changes not only in the behavior of animals but also in several aspects associated with GI motor function and sensitivity. These changes, although not associated with microscopic alterations of the gut wall, were sex-dependent and more evidently associated with the diet displaying the highest content in ω-6 PUFAs (especially linoleic acid).

Whether or not these changes may predispose to the development of a proper brain–gut axis disorder, particularly IBS, after longer exposure to these diets or after addition of other triggers (i.e., stress, infection) needs to be determined, as well as the specific impact of the different FAs in the diet of patients suffering from this kind of disorder.

## Figures and Tables

**Figure 1 nutrients-13-01864-f001:**
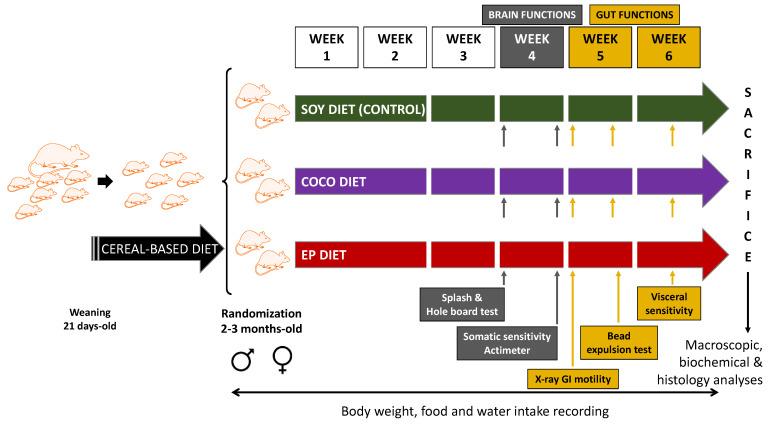
Study protocol. Rats were fed on a standard, cereal-based diet until the age of 2–3 months. Then, they were randomly allocated to three diet groups: SOY diet, i.e., the validated AIN-93G diet, whose only source of fat is soybean oil; COCO diet, i.e., AIN-93G diet enriched in coconut oil (rich in saturated fatty acids); and EP diet, i.e., diet enriched in evening primrose oil (rich in polyunsaturated fatty acids). SOY diet was the control diet of the study. Body weight, food and water intakes were recorded during the study. Brain functions were evaluated during week 4 of diet intervention using splash and hole-board tests, somatic sensitivity tests, and actimeter. Gut functions were estimated from week 5 to week 6 on the basis of X-ray motility analysis, colonic bead expulsion test, and visceral (colonic) sensitivity. After six weeks of diet intervention animals were sacrificed, and macroscopic, biochemical, and histology analyses were performed. Animals used for visceral sensitivity studies were not used for plasma/tissue collection.

**Figure 2 nutrients-13-01864-f002:**
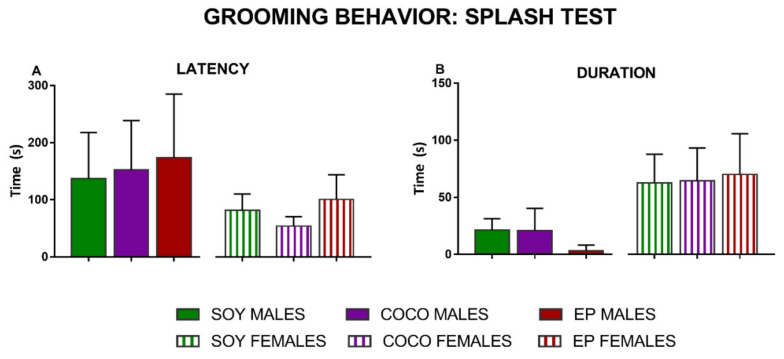
Effects of diet intervention on anhedonia (grooming behavior in the splash test) in the rat. Male and female rats were exposed to AIN-93G diet containing 7% of soybean oil (SOY), or to modified AIN-93G diet containing 3.5% of soybean oil supplemented with 3.5% of coconut oil (COCO) or with 3.5% of evening primrose oil (EP). Both latency (**A**) and duration (**B**) of grooming behavior after sucrose spraying (splash test) were recorded and analyzed in males and females separately. Data are expressed as mean ± S.D. (*n* = 6–10 animals per group). One-way ANOVA followed by Bonferroni’s multiple comparison test.

**Figure 3 nutrients-13-01864-f003:**
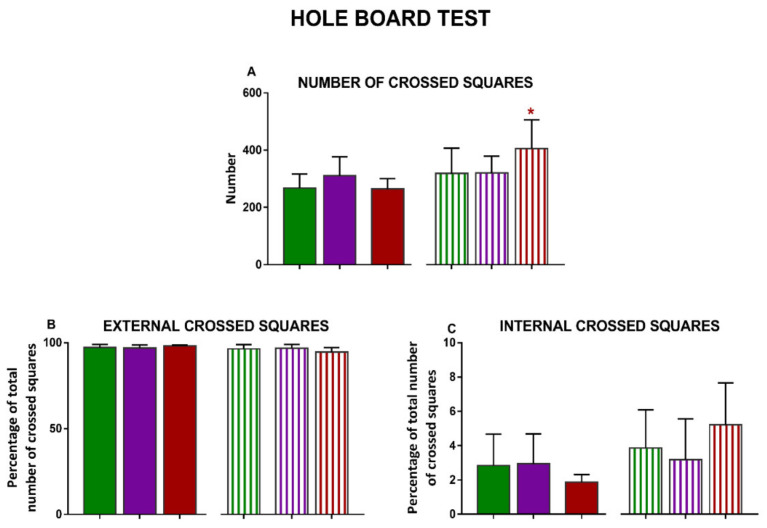
Effects of diet intervention on exploratory behavior (hole-board test) in the rat. Male and female rats were exposed to AIN-93G diet containing 7% of soybean oil (SOY), or to modified AIN-93G diet containing 3.5% of soybean oil supplemented with 3.5% of coconut oil (COCO) or with 3.5% of evening primrose oil (EP). Different parameters were evaluated during the test: total number of crossed squares (**A**), % of external (**B**) and internal (**C**) crossed squares, time spent at the holes (**D**), time standing on hindlimbs (**E**), time still (**F**), and number of feces expelled (**G**), analyzed in males and females separately. Data are expressed as means ± S.D. in normally distributed data and median and IQR when not normally distributed (*n* = 6–10 animals per group). One-way ANOVA followed by Bonferroni’s multiple comparison or Kruskal–Wallis test followed by Dunn´s multiple comparison test. * *p* < 0.05, ** *p* < 0.05 vs. SOY females.

**Figure 4 nutrients-13-01864-f004:**
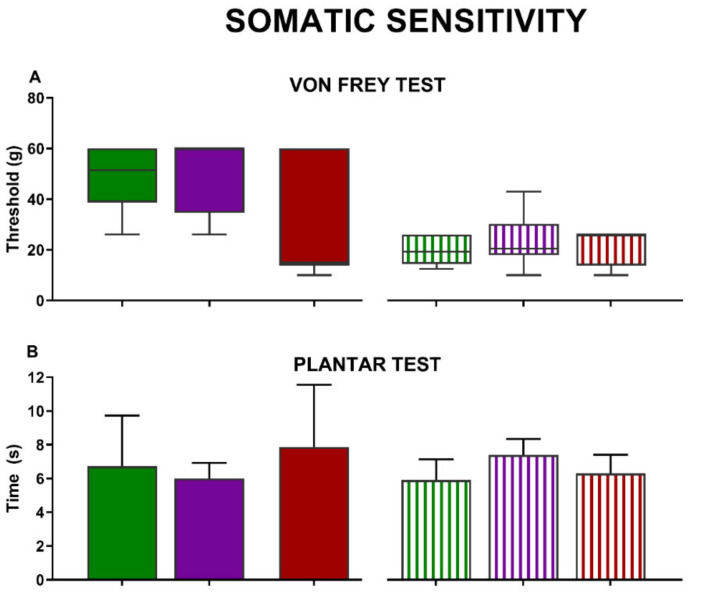
Effects of diet intervention on somatic sensitivity in the rat. Male and female rats were exposed to AIN-93G diet containing 7% of soybean oil (SOY), or to modified AIN-93G diet containing 3.5% of soybean oil supplemented with 3.5% of coconut oil (COCO) or with 3.5% of evening primrose oil (EP). Mechanical and thermal sensitivity were measured as the force or time necessary to elicit a nociceptive response in the von Frey (**A**), plantar test (**B**), and PAM test (**C**), analyzed in males and females separately. Data are presented as mean ± S.D. or median and I.Q.R, when not normally distributed (*n* = 6–13 per group). One-way ANOVA followed by Bonferroni’s multiple comparison or Kruskal–Wallis test followed by Dunn´s multiple comparison test.

**Figure 5 nutrients-13-01864-f005:**
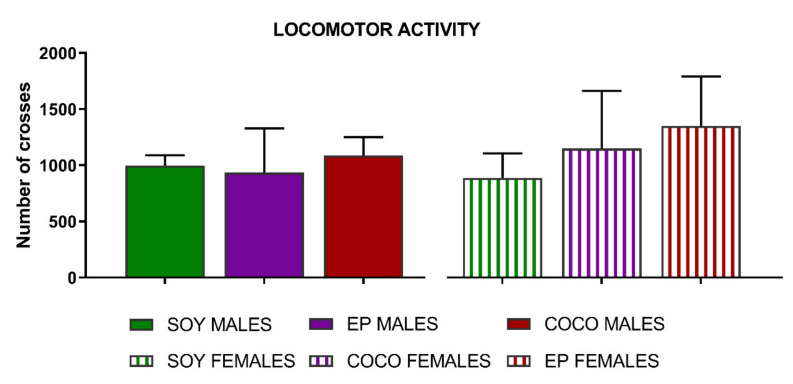
Effects of diet intervention on locomotor activity in the rat. Male and female rats were exposed to AIN-93G diet containing 7% of soybean oil (SOY), or to modified AIN-93G diet containing 3.5% of soybean oil supplemented with 3.5% of coconut oil (COCO) or with 3.5% of evening primrose oil (EP). Locomotor activity was measured as the total number of photobeam crosses in the actimeter and analyzed in males and females separately. Data are presented as mean ± S.D. (*n* = 6–13 per group). One-way ANOVA followed by Bonferroni’s multiple comparison.

**Figure 6 nutrients-13-01864-f006:**
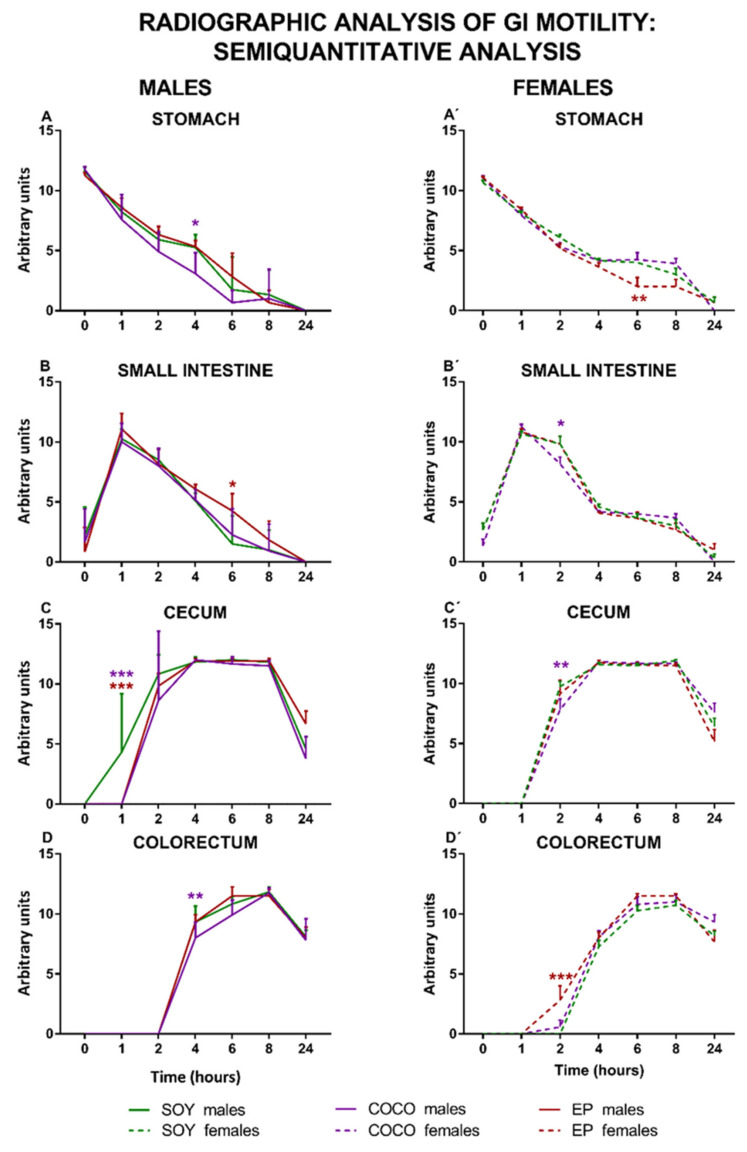
Effects of diet intervention on general gastrointestinal motility (semiquantitative radiographic analysis) in the rat. Male (**A**–**D**) and female (**A’**–**D’**) rats were exposed to AIN-93G diet containing 7% of soybean oil (SOY), or to modified AIN-93G diet containing 3.5% of soybean oil supplemented with 3.5% of coconut oil (COCO) or with 3.5% of evening primrose oil (EP). Figures represent the motility of stomach (**A**,**A’**), small intestine (**B**,**B’**), cecum (**C**,**C’**), and colorectum (**D**,**D’**) based on X-rays obtained 0, 1, 2, 4, 6, 8, and 24 h after contrast administration. Data were analyzed in males and females separately. Data are presented as mean ± S.D. (*n* = 6–12 animals per group) Two-way ANOVA followed by Bonferroni’s multiple comparison. * *p* < 0.05, ** *p* < 0.01, *** *p* < 0.001 vs. SOY males or SOY females.

**Figure 7 nutrients-13-01864-f007:**
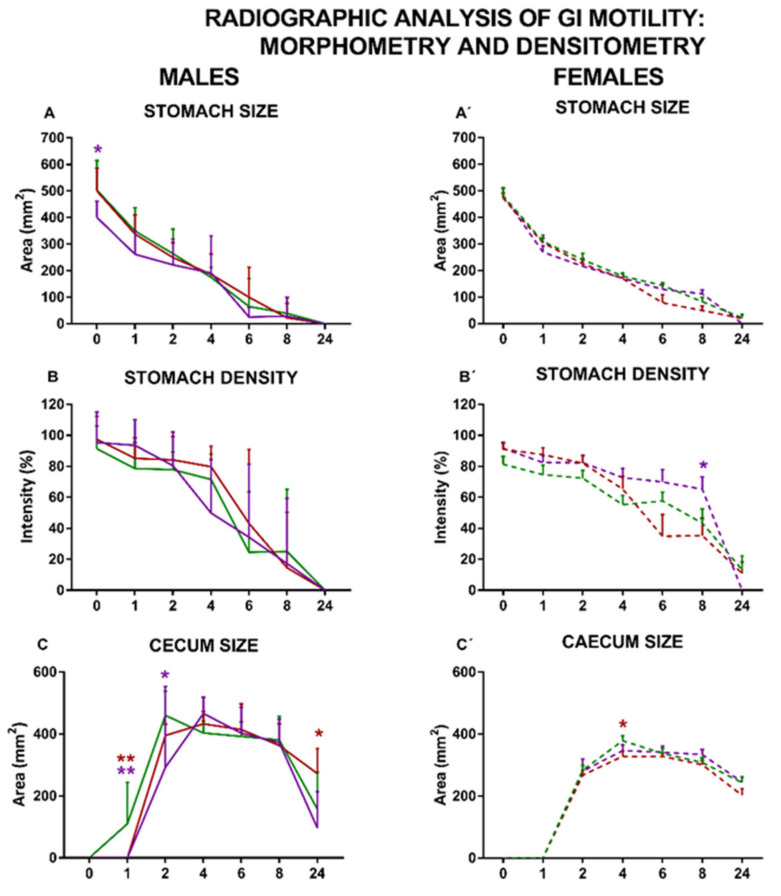
Effects of diet intervention on size and density of the stomach and cecum in the rat. Male (**A**–**D**) and female (**A’**–**D’**) rats were exposed to AIN-93G diet containing 7% of soybean oil (SOY), or to modified AIN-93G diet containing 3.5% of soybean oil supplemented with 3.5% of coconut oil (COCO) or with 3.5% of evening primrose oil (EP). Figures represent the size (**A**,**A’**) and density (**B**,**B’**) of stomach, and the size (**C**,**C’**) and density (**D**,**D’**) of cecum based on X-rays obtained 0, 1, 2, 4, 6, 8, and 24 h after contrast administration. Data were analyzed in males and females separately. Data are presented as mean ± S.D. (*n* = 6–12 animals per group). Two-way ANOVA followed by Bonferroni’s multiple comparison. * *p* < 0.05, ** *p* < 0.01 and *** *p* < 0.001 vs. SOY males or females.

**Figure 8 nutrients-13-01864-f008:**
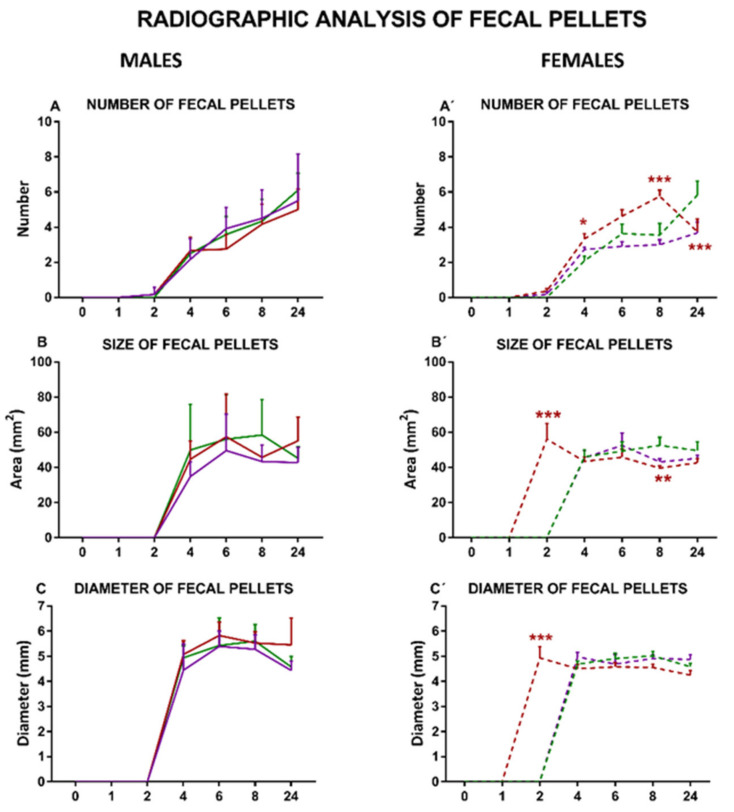
Effects of diet intervention on size and density of fecal pellets in the rat. Male (**A**–**D**) and female (**A’**–**D’**) rats were exposed to AIN-93G diet containing 7% of soybean oil (SOY), or to modified AIN-93G diet containing 3.5% of soybean oil supplemented with 3.5% of coconut oil (COCO) or with 3.5% of evening primrose oil (EP). Figures represent the number (**A**,**A’**), size (**B**,**B’**), diameter (**C**,**C’**), and density (**D**,**D’**) of fecal pellets based on X-rays obtained 0, 1, 2, 4, 6, 8, and 24 h after contrast administration. Data were analyzed in males and females separately. Data are presented as mean ± S.D. (*n* = 6–12 animals per group). Two-way ANOVA followed by Bonferroni’s multiple comparison. * *p* < 0.05, ** *p* < 0.01 and *** *p* < 0.001 vs. SOY males or females.

**Figure 9 nutrients-13-01864-f009:**
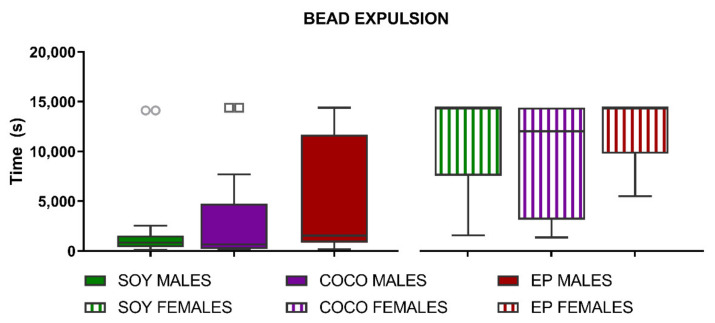
Effects of diet intervention on colonic bead expulsion in the rat. Male and female rats were exposed to AIN-93G diet containing 7% of soybean oil (SOY), or to modified AIN-93G diet containing 3.5% of soybean oil supplemented with 3.5% of coconut oil (COCO) or with 3.5% of evening primrose oil (EP). Colonic propulsion was measured as latency time after bead insertion, with a cut-off time of 14,400 s. Data were analyzed in males and females separately. Data are presented as median ± IQR (*n* = 6–8 animals per group). Kruskal–Wallis test was performed followed by Dunn´s multiple comparison test.

**Figure 10 nutrients-13-01864-f010:**
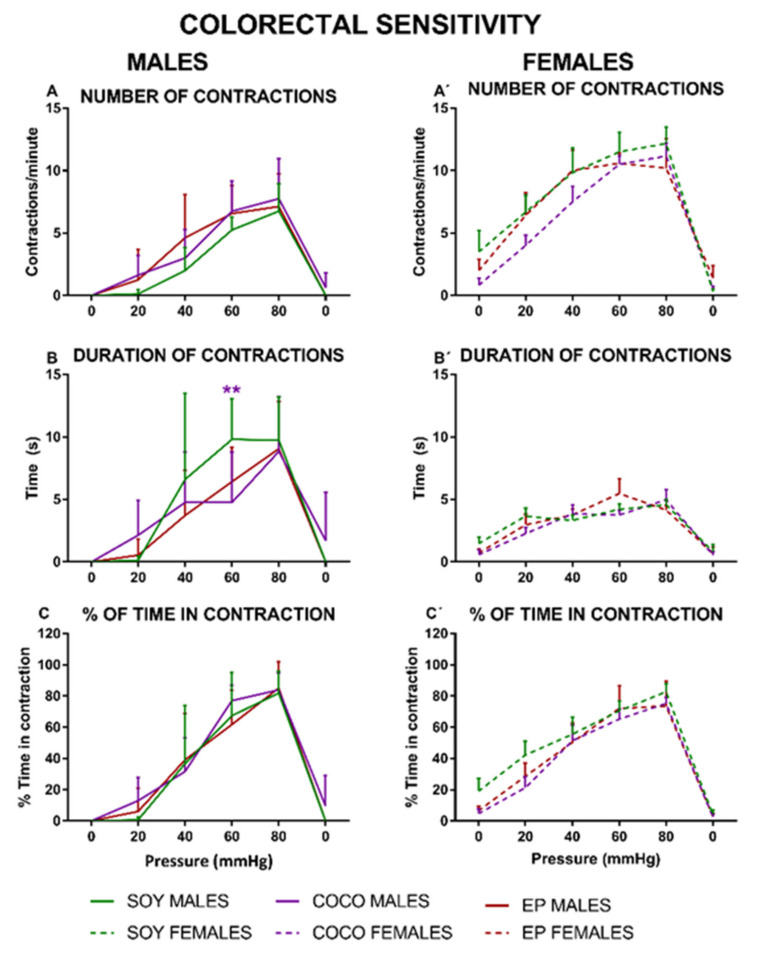
Effects of diet intervention on colonic sensitivity in the rat. Male (**A**–**C**) and female (**A’**–**C’**) rats were exposed to AIN-93G diet containing 7% of soybean oil (SOY), or to modified AIN-93G diet containing 3.5% of soybean oil supplemented with 3.5% of coconut oil (COCO) or with 3.5% of evening primrose oil (EP). Colonic sensitivity was measured as the number of abdominal contractions (**A**,**A’**), duration of the contractions (**B**,**B’**), and % of time in contraction (**C**,**C’**) in response to intracolonic mechanical phasic stimulation (0–80 mmHg, each stimulus lasting for 20 s, 3 stimuli per pressure interval, separated by 60 s). Data were analyzed in males and females, separately. Data are presented as mean ± S.D. (*n* = 6–8 per group). Two-way ANOVA followed by Bonferroni’s multiple comparison. ** *p* < 0.01 vs. SOY males.

**Table 1 nutrients-13-01864-t001:** Ingredients (**A**) and fatty acids (**B**) in diets.

**A**			
**Ingredients (%)**	**SOY**	**COCO**	**EP**
Corn starch	39.75	39.75	39.75
Casein (>85% of protein)	20.0	20.0	20.0
Maltodextrin	13.2	13.2	13.2
Sucrose	10.0	10.0	10.0
Soybean oil	7.0	3.5	3.5
Coconut oil	-	3.5	-
Evening primrose oil	-	-	3.5
Fiber (cellulose)	5.0	5.0	5.0
Mineral mix	3.5	3.5	3.5
Vitamin mix	1.0	1.0	1.0
L-cysteine	0.3	0.3	0.3
Choline Hydrogen Tartrate	0.25	0.25	0.25
Tertiary-Butylhydroquinone	0.001	0.001	0.001
**Total**	**100**	**100**	**100**
**Energy density (kcal/kg)**	**4030**	**4030**	**4030**
Energy from proteins (%)	20	20	20
Energy from lipids (%)	16	16	16
Energy from carbohydrates (%)	64	64	64
**B**			
**Type of Fatty Acid**	**SOY (%)**	**COCO (%) ***	**EP (%) ***
**SATURATED FAs**	**14**	**52.2**	**11.1**
C8:0 caprylic acid		3.55	
C10:0 capric acid		2.85	
C12:0 lauric acid		23.4	
C14:0 myristic acid		9.1	
C16:0 palmitic acid	10	9.9	8.16
C18:0 stearic acid	4	3.4	2.94
**MONOUNSATURATED** **LCFAs**	**23**	**14.9**	**14.97**
C18:1 n-9 oleic acid	23	14.9	14.97
**POLYUNSATURATED** **LCFAs**	**58**	**30**	**70.56**
C18:2 n-6 linoleic acid	51	26.4	62.44
C18:3 n-3 linolenic acid	7	3.55	3.5
C18:3 n-6 Gamma-linolenic acid		0.05	4.62
**Others #**	**5**	**2.9**	**3.38**
Total	100	100	100

Abbreviations: SOY—AIN-93G diet containing 7% of soybean oil; COCO—modified AIN-93G diet containing 3.5% of soybean oil and supplemented with 3.5% of virgin coconut oil; EP—modified AIN-93G diet containing 3.5% of soybean oil and supplemented with 3.5% of evening primrose oil; FAs—fatty acids; LCFAs—long-chain fatty acids. *—% Values have been calculated according to the information specified by the diet providers in the data sheets of the different diets. #—Corresponds to non-specified FAs.

**Table 2 nutrients-13-01864-t002:** Effects of diet intervention on gut wall architecture in the rat.

	Sex	SOY	COCO	EP
**Ileal damage (a. u.)**	Males	1.83 ± 0.98	2.83 ± 1.7	3.00 ± 0.89
	Females	2.00 ± 1.1	1.83 ± 0.75	2.33 ± 1.21
**Colonic damage (a. u.)**	Males	3.83 ± 1.94	3.66 ± 0.82	4.16 ± 1.33
	Females	3.16 ± 0.98	3.33 ± 1.03	3.66 ± 1.21
**Colonic longitudinal** **muscle thickness (µm)**	Males	47.40 ± 19.15	40.60 ± 3.82	44.50 ± 9.27
	Females	41.10 ± 8.99	43.80 ± 13.75	42.70 ± 14.7
**Colonic circular** **muscle thickness (µm)**	Males	185 ± 46.59	184 ± 27.4	209 ± 44.49
	Females	183 ± 32.9	192 ± 60.13	209 ± 91
**Number of mast cells (*n*)**	Males	7.01 ± 3.86	8.10 ± 3.67	7.78 ± 2.23
	Females	6.46 ± 4.07	7.05 ± 4.11	7.80 ± 2.81

Male and female rats were exposed to AIN-93G diet containing 7% of soybean oil (SOY), or to modified AIN-93G diet containing 3.5% of soybean oil supplemented with 3.5% of coconut oil (COCO) or with 3.5% of evening primrose oil (EP). Damage quantification on ileal and colonic tissues, muscular layer thickness, and number of mast cells infiltrating colonic submucosa per microscopic field (20×) were recorded. The maximum damage scores for the ileum and colon are 9 and 13, respectively. Data were analyzed in males and females separately. Data are expressed as the means ± S.D. (*n* = 6 animals per group). One-way ANOVA followed by Bonferroni´s multiple comparison test. Abbreviations: a. u., arbitrary units.

**Table 3 nutrients-13-01864-t003:** Summary of relevant effects of changes in dietary fatty acid composition on the brain–gut axis functions.

		Brain	Gut
Sex	Diet	Exploration & Locomotion	Grooming	Colonic Sensitivity	Transit	Content Density
S	SI	C	CR	C	FP
** 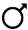 **	**COCO**			X	X		X	X		
**EP**		<X			X	X			
** 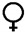 **	**COCO**			<X		X	X		X	X
**EP**	X			X			X	X	X

Compared with AIN-93G (termed as control or SOY throughout the text) diet containing 7% soybean oil, male and female rats exposed to modified AIN-93G diet containing 3.5% of soybean oil supplemented with 3.5% of coconut oil (COCO) or with 3.5% of evening primrose oil (EP) displayed some alterations in brain (emotional behavior, exploration and locomotion, grooming) and gut functions (transit, colonic sensitivity, content density) that were significant (X) or almost significant (<X). Abbreviations: C, cecum; CR, colorectum; FP, fecal pellets (within the colorectum); S, stomach; SI, small intestine.

## Data Availability

The datasets generated and/or analyzed during the current study are available from the corresponding author on reasonable request.

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
