# Peer review of "Changes in Fatty Acid Dietary Profile Affect the Brain–Gut Axis Functions of Healthy Young Adult Rats in a Sex-Dependent Manner"

_nutrients, 2021, doi:10.3390/nu13061864_

Round 1
Reviewer 1 Report
The authors present a large and complex rodent trial comparing four
randomised diets and assessing a broad variety of behavioral and somatic
outcomes. The selected comparisons are relevant for several clinical
questions, showing sex-dependent effects of certain diets. The rationale for this particular experiment should be part of the abstract.
Methods are explained well and conducted by the state of the art.
However, there is no indication of a power calculation, of blinding or randomisation of animals for the respective test sessions. Please clarify.
Results:
Chapter 3.4. seems to contain a ceiling effect at 15000 s; this should be mentioned in this chapter already.
In order to improve the readability of the manuscript, I recommend summarizing all findings of behavioral and GI functional experiments in a final master table, containing all comparisons, outcomes and significant findings (after correction for multiple testing).
Discussion:
The very voluminous discussion addresses all relevant aspects of the conducted rodent trial, however, given the overwhelming multitude of outcomes and significant risk of spurious, false-positive effects, it needs to be adapted to the yet missing statistical correction for multiple comparisons. In general, the discussion lacks mentioning limitations of the study. A major flaw - not to be changed anymore - is the problem of single measurements after the intervention without measuring the outcome variables before the intervention. Thus, comparisons between the groups can be affected by random differences at baseline.
Reviewer 2 Report
The authors are aiming to evaluate the effect of diet with different FA composition on rat's behavior and gut motility and histology. Based on Fig. 2B males in group EP had a higher food intake than males in SOY and COCO groups. This fact invalid the study objective: authors cannot know if any observed and reported effect is due to an effect of the diet itself (aka. FA composition) OR a higher energy intake. In fact, most of the significant results are observed only in EP males. The results from Table 2 further supports a higher food intake; EP males had a sign increased weight. This is a fatal flow in the study.
Reviewer 3 Report
The manuscript by Jacenik et al. evaluated e the effects produced by three purified diets with different FA composition on the brain-gut axis in both male and female rats. The authors suggest that changes in FA composition may facilitate the development of brain-gut axis alterations in a sex-dependent manner. The study is comprehensive and generally support the authors' conclusions. These findings may benefit from some additional clarification, as detailed below.
Comments
- Did the authors use a control diet?
- Did the authors observe changes in serum triglycerides and cholesterol concentrations?
- It should be very interesting if the authors administer an oleic acid-rich diet.
- The manuscript should be edited to correct contextual and layout errors.
Round 2
Reviewer 1 Report
The authors have thoroughly and sufficiently addressed all points of criticism.
Reviewer 2 Report
This study has not been approved as it as designed incorrectly. Given the aim of the study, it is inappropiate to adjust the body weight by the food intake as this is not a linear correlation. This INVALIDATES the entire study.
Reviewer 3 Report
//
This manuscript is a resubmission of an earlier submission. The following is a list of the peer review reports and author responses from that submission.
Round 1
Reviewer 1 Report
This study was to address the influence of dietary factors, namely fiber and fatty acids, on how they influence the gut-brain axis. The introduction was fairly vague in terms of any information surrounding this area of research which has been ongoing for quite some time. Rather, they only describe briefly how these dietary factors influence immune function and metabolic disease. They reference a previous publication which looked at how dietary fiber and fatty acids lower gastro-intestinal motility, but have no impact on cardiovascular parameters. They are extending these findings looking at male and female rats.
It is uncertain why these investigators describe this study as a way to evaluate the influence of dietary fiber on behavior and GI function because they use completely different diets for this part of the objective – The control diet (standard diet) is very high in fiber, and much higher than the other low fat diets, but the problem is that this diet is a grain-based diet, which has many other differences from the 3 other lower fiber purified diets, making data interpretation impossible. In order for them to understand how fiber influences their data, this would require a group fed a purified diet based on the other 3 diets where they modify fiber only. In their diet, the fibers in the grain-based diet are undefined and complex and coming from multiple grain sources. The use of this grain-based diet as a ‘high fiber diet’ is really a fatal flaw and any conclusions mentioned about how fiber influences their parameters (i.e. comparisons between “CTRL” and “LFDs” are impossible due to the many other factors in these diets that differ.
This control diet can’t be used for the purified diets in order to study how fatty acids influence their parameters of interest again for the same reason – the control diet contains different fats, but there are other differences. For the fatty acid comparisions, SOY diet could be the control diet and then compare effects by COCO and EP diets to those fed the control. However, EP oil, like soybean oil, contains a high concentration of linoleic acid as the experimental diets, they should do a better job clarifying the decision to use this EP oil. If they are to add only ½ of the test fat as the experimental fat and the other ½ as soybean oil, then this diet with EP oil will have around 67% linoleic acid (given 54% in soybean oil and 80% in EP oil), so it would have been better to reduce the soybean oil in this diet to perhaps 1% and have 6% of the test oil. To this end, there should have been a fatty acid profile given for the final profile in the COCO and EP diets, which would have reflected a different profile than the original oils.
We are led to believe that this study will be on the etiopathology of gut-brain axis disorders. They perform numerous sensitivity tests (both somatic and visceral), which I believe could be of interest, but while there are some differences noted (some cases gender, some diet), one is unsure of whether these are significant given the improper control diet. Similarly, changes in the gut histology and other GI functional tests performed also are difficult to really understand their significance.
The discussion is difficult to follow. For example, I didn’t follow well when discussing about female mice results and extending their findings to obesity in pre-menopausal and post-menopausal women. Because this is not an obesity study, it seems irrelevant to bring their findings up in this context. After all, the LFDs did not cause obesity.
Conclusions are unfounded. No functional gastrointestinal disorders were studied, yet there is a claim that diet may be used as a supportive therapy in the context of functional gastrointestinal disorders. Again, it really must be emphasized that the control diet is not appropriate and no such conclusions can be made with this study.
We end up knowing little about how diet composition affects the parameters studied. Interestingly, while the title of this paper says this is about the influence of fiber in the introduction and methods, no mention of it is found in the conclusions. The control diet was inappropriate.
Author Response
Dear reviewer,
Thank you very much for your comments, that have allowed us to greatly improve our manuscript.
This study was to address the influence of dietary factors, namely fiber and fatty acids, on how they influence the gut-brain axis. The introduction was fairly vague in terms of any information surrounding this area of research which has been ongoing for quite some time. Rather, they only describe briefly how these dietary factors influence immune function and metabolic disease. They reference a previous publication which looked at how dietary fiber and fatty acids lower gastro-intestinal motility, but have no impact on cardiovascular parameters. They are extending these findings looking at male and female rats.
ANSWER: As suggested by the reviewer, we have thoroughly revised the Introduction to make it more focused on the brain-gut axis and FGID, including IBS. We think the aims of our study and their connection with these disorders are now much clearer and hope the Introduction is now acceptable.
It is uncertain why these investigators describe this study as a way to evaluate the influence of dietary fiber on behavior and GI function because they use completely different diets for this part of the objective – The control diet (standard diet) is very high in fiber, and much higher than the other low fat diets, but the problem is that this diet is a grain-based diet, which has many other differences from the 3 other lower fiber purified diets, making data interpretation impossible. In order for them to understand how fiber influences their data, this would require a group fed a purified diet based on the other 3 diets where they modify fiber only. In their diet, the fibers in the grain-based diet are undefined and complex and coming from multiple grain sources. The use of this grain-based diet as a ‘high fiber diet’ is really a fatal flaw and any conclusions mentioned about how fiber influences their parameters (i.e. comparisons between “CTRL” and “LFDs” are impossible due to the many other factors in these diets that differ.
This control diet can’t be used for the purified diets in order to study how fatty acids influence their parameters of interest again for the same reason – the control diet contains different fats, but there are other differences. For the fatty acid comparisions, SOY diet could be the control diet and then compare effects by COCO and EP diets to those fed the control. However, EP oil, like soybean oil, contains a high concentration of linoleic acid as the experimental diets, they should do a better job clarifying the decision to use this EP oil. If they are to add only ½ of the test fat as the experimental fat and the other ½ as soybean oil, then this diet with EP oil will have around 67% linoleic acid (given 54% in soybean oil and 80% in EP oil), so it would have been better to reduce the soybean oil in this diet to perhaps 1% and have 6% of the test oil. To this end, there should have been a fatty acid profile given for the final profile in the COCO and EP diets, which would have reflected a different profile than the original oils.
ANSWER: Thank you very much for pointing these issues out. We really appreciate the time devoted to thoroughly review our manuscript and the good suggestions for its improvement and for future work. In particular, we highly appreciate the suggestions posed by the reviewer about the need to evaluate other diets, particularly a purified diet, with both fiber and fat total percentages similar to those in CTRL diet. We have thoroughly revised our manuscript, particularly the introduction and aims, and we hope the reasoning under our diet choices are now clearer.
As mentioned by the reviewer, we used a standard, undefined, grain-based diet as reference for the whole study. Standard laboratory diets like our CTRL diet are used everywhere and are probably much more translational (humans normally feed on non-define diets). Purified diets have a particular composition and the advantage of using them is that studies from different laboratories or at different moments will be more comparable. One such purified diet is AIN-93M. This diet was compared with a standard cereal-based diet in a two-year long study and it did not modify rat growth and survival under normal conditions; however, under conditions of caloric restriction, AIN-93M diet performed worse than the non-purified diet, and the authors suggested that cereal-based diets may provide the animals with elements that are missing in defined diets (PMID: 11773515, doi: 10.1093/jn/132.1.101). However, irrespective of the caloric conditions, both diets were equivalent in the short term, at least regarding normal growth and general health of the animals (PMID: 11773515, doi: 10.1093/jn/132.1.101).
On the other hand, AIN-93M diet was modified to develop another kind of diet, with a higher amount of fat, more suitable for very young, growing rodents (PMID: 9164249, doi: 10.1093/jn/127.5.838S). This new diet is AIN-93G, the same purified diet that we used in our study, whose fat source is soybean oil (that is why we called it “SOY” diet). Importantly, compared with AIN-93M and CTRL diets, AIN-93G diet has 7% of fat and almost double energy density, but a similar (balanced) fatty acid composition. Moreover, although the increase in % of fat in AIN-93G/SOY diet is accompanied by a reduction in fiber, we agree with the reviewer in that putting the focus of our study on fiber makes it very confusing. Instead, we believe that the focus should be put on total fat proportion and its composition. Thus, we considered AIN-93G diet the ideal one to address two aims (which we believe are much more clearly explained in the new version of our manuscript):
- AIM 1: to determine the impact of increasing the proportion of fat on the parameters of interest related with the brain-gut axis; for this, we evaluated the effects of AIN-93G/SOY diet in both males and females, and used CTRL diet as reference (although other differences may occur and should be evaluated in the future, the most important one for the present study is the total amount of fat, not the fatty acid or fiber composition)
- AIM 2: to determine the impact, on the same brain-gut axis parameters, of changing the type of fatty acids in fat; in this case, AIN-93G/SOY diet was used as reference and the effect of changing half of the soybean oil for coconut oil (particularly rich in saturated fatty acids) or for evening primrose oil (particularly rich in polyunsaturated fatty acids) was studied in each sex (these modified diets were called COCO and EP diets, respectively); the three diets used to address this aim are high in fat proportion, and therefore are termed high-fat diets along the text; our previous experience (PMID: 31145538, doi: 10.1111/nmo.13651) suggested that, even if relatively small, the modifications in the diets used here had an impact on the brain-gut axis that was worth being more deeply studied, although we agree in that other coconut or evening primrose oil compositions in COCO and EP diets (as those suggested by the reviewer) should be tested and we will very likely address this in our future studies)
We have made these considerations in the different sections of the manuscript and hope it is now acceptable.
We are led to believe that this study will be on the etiopathology of gut-brain axis disorders. They perform numerous sensitivity tests (both somatic and visceral), which I believe could be of interest, but while there are some differences noted (some cases gender, some diet), one is unsure of whether these are significant given the improper control diet. Similarly, changes in the gut histology and other GI functional tests performed also are difficult to really understand their significance.
The discussion is difficult to follow. For example, I didn’t follow well when discussing about female mice results and extending their findings to obesity in pre-menopausal and post-menopausal women. Because this is not an obesity study, it seems irrelevant to bring their findings up in this context. After all, the LFDs did not cause obesity.
Conclusions are unfounded. No functional gastrointestinal disorders were studied, yet there is a claim that diet may be used as a supportive therapy in the context of functional gastrointestinal disorders. Again, it really must be emphasized that the control diet is not appropriate and no such conclusions can be made with this study.
We end up knowing little about how diet composition affects the parameters studied. Interestingly, while the title of this paper says this is about the influence of fiber in the introduction and methods, no mention of it is found in the conclusions. The control diet was inappropriate.
ANSWER: Thank you very much for your interest in our results and for your comments regarding the focus of our study, that we believe have allowed us to greatly improve our manuscript. As previously stated, we have thoroughly revised the Introduction, Results and Discussion to make the manuscript more focused on the brain-gut axis function (and dysfunction). In addition, we have prepared a summary figure that we believe is very useful to relate and integrate our findings. The Discussion (and the summary figure) is organized now in three main issues that offer explanations in a progressive way: the impact of sex; the impact of a higher amount of dietary fat; the impact of changing fatty acid composition. In addition, we have reviewed the Conclusions and think they are also more focused and better related with our aims now. We hope the new version of our manuscript is now acceptable.
Reviewer 2 Report
The authors present a large and complex rodent trial comparing four randomised diets and assessing a broad variety of behavioral and somatic outcomes. The selected comparisons are relevant for several clinical questions, showing sex-dependent effects of certain diets.
The rationale for this particular experiment - as given in the introduction - is kept mainly unclear until the discussion. There is no explanation, why effects of sex should be investigated with respect to these four particular diets and the enormous number of very diverse outcomes.
For the entire manuscript, "gender" should be replaced by "sex" for rats or "gender and sex", if both animals and humans are referred to in the same context.
Methods are explained well and conducted by the state of the art. However, there is no indication of a power calculation or of blinding (apart from the hole-board test) or randomisation of animals for the respective test sessions. Please clarify.
Table 1 needs to be re-ordered, starting with macronutrients, their subspecies, and ending with non-caloric micronutrients.
Please clarify, if normal distribution of the data was assured; which tests were used for non-normally distributed data. Please use mean+SD for ND data, and median+IQR for nND data. SEM is not a statistically suitable measure of variance for this kind of study. Given the enormous number of tests without pinpointed hypotheses, correction for multiple comparisons is required.
Results:
Figure 2 is redundant to Fig. 3; differences by sex could be demonstrated by overarching bars between the respective graphs. This applies to the later diagrams as well. In order to improve the readability of the manuscript, I recommend summarizing all findings in a final master table, containing all comparisons, outcomes and significant findings (after correction for multiple testing).
Discussion:
The reference to IBS comes rather surprising to the reader's attention. The first paragraphs might better serve as part of the introduction, not the discussion. The very voluminous discussion addresses all relevant aspects of the conducted rodent trial, however, given the overwhelming multitude of outcomes and significant risk of spurious, false-positive effects, it needs to be adapted to the yet missing statistical correction for multiple comparisons. A final review of the discussion is therefore only possible after the necessary statistical revision.
Author Response
Dear reviewer,
Thank you very much for your comments, that have allowed us to greatly improve our manuscript.
The authors present a large and complex rodent trial comparing four randomised diets and assessing a broad variety of behavioral and somatic outcomes. The selected comparisons are relevant for several clinical questions, showing sex-dependent effects of certain diets. The rationale for this particular experiment - as given in the introduction - is kept mainly unclear until the discussion. There is no explanation, why effects of sex should be investigated with respect to these four particular diets and the enormous number of very diverse outcomes.
ANSWER: Thank you very much for your interest in our study and your comments regarding sex-dependent effects.According to the Reviewer’s suggestion, the Introduction has been rearranged and in our opinion it is more focused now on functional gastrointestinal disorders and the impact of diet on them. Furthermore, the relevance of gender-related differences in patients with functional gastrointestinal disorders and their relationship with diet is now presented in this section with updated literature. We hope this is now acceptable.
For the entire manuscript, "gender" should be replaced by "sex" for rats or "gender and sex", if both animals and humans are referred to in the same context.
ANSWER: Thank you very much for pointing this out. According to the Reviewer’s suggestion, the terms “sex” and “gender” have been now used for rats and humans, respectively.
Methods are explained well and conducted by the state of the art. However, there is no indication of a power calculation or of blinding (apart from the hole-board test) or randomisation of animals for the respective test sessions. Please clarify.
ANSWER: Thank you very much for your comment on our methods. According to the Reviewer’s suggestion, we clearly state now that “All experimental procedures were performed and analyzed blindly with respect to diet type exposure of the animals” in line 138. In addition, the cages chosen for each test were randomly allocated (visceral sensitivity or tissue obtention), this has been added in line 153. Finally, based on previous studies of our group, we calculated the sample size needed for the study with G*power, with an α error 0.05 and power = 0.8, and we considered an effect size of 0.25 as relevant. The result suggested that 6 animals were needed in each experimental group. Thus, we considered 6 as the minimum number of animals per experimental group (allocated to the different tests) necessary to obtain reliable results in our study. We have explained this in the corresponding section (statistical analysis). We hope this is now acceptable.
Table 1 needs to be re-ordered, starting with macronutrients, their subspecies, and ending with non-caloric micronutrients.
ANSWER: According to the reviewer’s suggestion Table 1 was re-ordered starting with macronutrients, their subspecies and non-caloric micronutrients.
Please clarify, if normal distribution of the data was assured; which tests were used for non-normally distributed data. Please use mean+SD for ND data, and median+IQR for nND data. SEM is not a statistically suitable measure of variance for this kind of study. Given the enormous number of tests without pinpointed hypotheses, correction for multiple comparisons is required.
ANSWER: Thank you very much for your comment. The data have been first analyzed using the Shapiro-Wilk normality test to determine if they were normally distributed. Additionally, we have used Kruskall-Wallis test followed by Dunn´s multiple comparison test to compare not normally distributed data. Regarding normally distributed data, we have used One- or Two-way ANOVA and modified the previous pot-hoc test to Bonferroni’s multiple comparisons post-hoc test. This information has been added to the statistical analysis section. Most data were distributed normally and are now represented using SD instead of SEM, as suggested by the reviewer. Likewise, the scarce data that were not distributed normally are now presented as median+IQR. We hope this is now acceptable.
Results: Figure 2 is redundant to Fig. 3; differences by sex could be demonstrated by overarching bars between the respective graphs. This applies to the later diagrams as well.
ANSWER: Thank you very much for these useful comments. According to the suggestions performed by reviewer 1, we have clarified the aims of our study and have analyzed all data again accordingly, with new figures to better illustrate our results (including those that are now shown as Supplementary Material). We believe the new way we present our data makes our results more understandable and more focused. We hope the reviewer finds this appropriate.
In order to improve the readability of the manuscript, I recommend summarizing all findings in a final master table, containing all comparisons, outcomes and significant findings (after correction for multiple testing).
ANSWER: We have added a new figure (Figure 11) summarizing the main results obtained and thank the reviewer for this idea. We believe this figure is really helpful to increase the understanding of our findings.
Discussion: The reference to IBS comes rather surprising to the reader's attention. The first paragraphs might better serve as part of the introduction, not the discussion. The very voluminous discussion addresses all relevant aspects of the conducted rodent trial, however, given the overwhelming multitude of outcomes and significant risk of spurious, false-positive effects, it needs to be adapted to the yet missing statistical correction for multiple comparisons. A final review of the discussion is therefore only possible after the necessary statistical revision.
ANSWER: Thank you very much for these useful ideas. As suggested by the reviewer, we have now rephrased the Introduction and is much more focused on the functional gastrointestinal disorders, particularly irritable bowel syndrome, as well as the different factors relevant to their development and clinical impact (sex, diet, brain-gut axis). Moreover, as mentioned above, we have reviewed our statistical analysis and our results are now organized according to our two aims. The Discussion section has benefited from all the improvements made in the previous sections in two ways: it is more focused, and also much shorter. We hope this is now acceptable.
Reviewer 3 Report
An article on: Diet composition of fiber and fatty acids alters rat behavior and gastrointestinal tract function in a sex-dependent manner was presented for review. The publication is very interesting in terms of application and valuable in terms of content, but it was not possible to fully emphasize all the elements in the content of the work. Perhaps too many analyzes were carried out, which blurred the final picture.
The study focuses on the effects of CTRL and LFD (Soy, Coco, EP) diets on behavioral responses and gastrointestinal motility, as well as visceral pain in male and female rats. It seems that the topic of the work should be extended to the aspect related to pain, because a large part of the work is about it.
Other detailed comments are provided below:
[111] Why did the experiment last 6 weeks? What factors determined this specific time window?
[117] Was the isocaloric value of the doses maintained with varying composition for all 4 groups of the rats? Did the diet differ in weight, structure and volume for specific doses? To be able to demonstrate correlation, it is necessary to remember that physical paraments (of food) are often crucial (for humans - only?)
[121] Whether total dietary fiber or water-soluble fiber was tested, as it is not clear from Table 1.A.
[121] Are LFDs averaged data from all trials (Soy, Coco, EP)?
[310] When comparing CTRL and LFDs diet statistics, the LFDs values should be included into the table to make it available for data comparison to make it available for comparison.
[592] What is the result most likely from? Which diet components are critical since polyunsaturated are responsible for EP = 80%. Can the results be justified with evening primrose oil supplementation only?
[611] Did the rats used in the study suffer from IBS? If they were not affected by the condition, is it justified to apply some of the test results to IBS patients?
[710] Are there any known differences in the modulation of abdominal pain between men and women by diet? Is ref. 82 a correct source in this matter?
[745] Here shall be more to elaborate on what type of dietary fiber and what daily dose is recommended as well as what the preferred fatty acid composition should be for IBS patients.
[835] Based on the presented research results, these conclusions remain unfounded.
[838] Diet is certainly the main element in the prevention of FGID, however this is not entirely clear from the results presented in this article. In addition, the summary should include recommendations on which food ingredients are particularly important in relieving the symptoms of IBS and other pain-related diseases. Although the direction is right, most of the work is based on assumptions to the applicability on human.
More focus shall be paid to the development of the methodology of research during tests on rats in terms of effectiveness and reliability of expected results, before making conclusions relating to IBS diet correlations on human. Continuation of human clinical trials is imperative to confirm the findings presented in the study.
[866] (.)
[868) (.)
[895] (,/.)
[909] (bold year)
[080] (bold year)
Author Response
Dear reviewer,
Thank you very much for your comments, which have allowed us to greatly improve our manuscript.
An article on: Diet composition of fiber and fatty acids alters rat behavior and gastrointestinal tract function in a sex-dependent manner was presented for review. The publication is very interesting in terms of application and valuable in terms of content, but it was not possible to fully emphasize all the elements in the content of the work. Perhaps too many analyzes were carried out, which blurred the final picture. The study focuses on the effects of CTRL and LFD (Soy, Coco, EP) diets on behavioral responses and gastrointestinal motility, as well as visceral pain in male and female rats. It seems that the topic of the work should be extended to the aspect related to pain, because a large part of the work is about it.
ANSWER: Thank you very much for your positive consideration towards our manuscript. According to your comments and those posed by the other reviewers, we have reviewed and almost in full rephrased the Introduction, Results and Discussion, as well as the Title and the Abstract. By doing this, we believe we have better focused the manuscript on our aims. We have also made a figure that summarizes our main findings (Figure 11) which shows that our results may have clinical application. The aspect of pain has been reviewed and is now discussed in relationship with the impact of sex, the impact of changing the amount of dietary fat and the impact of changing fatty acid composition in dietary fat. We hope the new version of our manuscript is now acceptable.
Other detailed comments are provided below:
[111] Why did the experiment last 6 weeks? What factors determined this specific time window?
ANSWER: We performed our animal study based on our previous original article (PMID: 31145538, doi: 10.1111/nmo.13651) where we were able to document that the short-term diet intervention, i.e., 4 weeks of treatment, may affect gastrointestinal and colonic motility as well as visceral sensitivity in male rats. Nevertheless, due to the addition of behavioral studies in this case we decided to extend treatment period to 6 weeks, to allow for resting periods in between procedures (the study protocol is shown in Figure 1). Indeed, the relatively short-term exposure to SOY, COCO and EP diets were already associated to multiple alterations in the studied parameters. In future studies, it will be interesting to determine if longer diet exposure may lead to further changes.
[117] Was the isocaloric value of the doses maintained with varying composition for all 4 groups of the rats? Did the diet differ in weight, structure and volume for specific doses? To be able to demonstrate correlation, it is necessary to remember that physical paraments (of food) are often crucial (for humans - only?)
ANSWER: As explained above and better explained now in the text, SOY, COCO and EP diets are characterized by the same percentage of ingredients and energy balance, but as shown in Table 1.B., each diet has a different % of fatty acid types. Additional statements explaining the percentage of energy density and percentage of ingredients available in each diet were added under Table 1. A. and along the text, where this information was relevant for understanding. We hope this has now been clarified.
[121] Whether total dietary fiber or water-soluble fiber was tested, as it is not clear from Table 1.A.
ANSWER: Thank you very much for your comment. In table 1.A. we provided the available information regarding fiber present in our diets. This information is indeed scarce, particularly for our undefined, cereal-based CTRL diet. In that table you can clearly see the proportion of insoluble fiber, but we cannot say much about soluble fiber (although it may be present, at least in CTRL diet). In any case, as explained above, the aims of the manuscript have been readjusted and now it is more focused on the effects of changing the total amount of fat and its fatty acid composition (amount and composition of fat seems to be better controlled for in these particular diets and make comparisons easier, although we acknowledge this is only a starting point for future studies in which other dietary components, including fiber and its types, may be specifically studied). We hope this is satisfactory.
[121] Are LFDs averaged data from all trials (Soy, Coco, EP)?
ANSWER: From your question, we realized that we had probably offered the information in a bit confusing way. Our experimental diets (i.e., SOY, COCO and EP diets) were considered separately as independent diets in all the experiments presented in our manuscript. As explained above and better explained now in the text, SOY, COCO and EP diets are characterized by the same percentage of ingredients and energy balance, but as shown in Table 1.B., each diet has a different % of fatty acid types. Additional statements explaining the percentage of energy density and percentage of ingredients available in each diet were added under Table 1. A. and along the text, where this information was relevant for understanding.
[310] When comparing CTRL and LFDs diet statistics, the LFDs values should be included into the table to make it available for data comparison to make it available for comparison.
ANSWER: As previously mentioned, we have rephrased our aims and reanalyzed our data so that we could compare the effects of CTRL and SOY diets first (AIM 1: to determine the impact of increasing fat proportion), and the effects of SOY, COCO and EP diets thereafter (AIM 2: to determine the impact of changing fatty acid composition). Therefore, the results for each diet on animals of each sex, are individualized. We hope this is clearer now.
[592] What is the result most likely from? Which diet components are critical since polyunsaturated are responsible for EP = 80%. Can the results be justified with evening primrose oil supplementation only?
ANSWER: Thank you very much for your comment. In the revised version of our manuscript, we have focused on the effects more related with the brain-gut axis. In these regards, the most interesting effects produced by COCO and EP diets relate with the decrease of SOY-induced hypersensitivity to intracolonic mechanical stimulation in male rats. Unfortunately, at this stage we cannot be sure of the specific molecule/s responsible for the effects obtained or the exact mechanisms involved, but we can consider the present results as an interesting starting point for more specific future studies that will address those aims.
[611] Did the rats used in the study suffer from IBS? If they were not affected by the condition, is it justified to apply some of the test results to IBS patients?
ANSWER: Thank you very much for your interesting comment. IBS is basically characterized by abdominal pain and altered defecation; in addition, altered emotional responses (depression, anxiety) are often associated; histologically, there is no patent signs of inflammation, but low-grade inflammation has been suggested to occur, mainly in relationship with increased numbers of mast cells. Our female rats displayed hypersensitivity to somatic and visceral stimuli, as well as reduced colonic propulsion; in addition, when exposed to SOY, their grooming behavior was reduced. Thus, these SOY-exposed females could be considered as a model of IBS with constipation. In contrast, we observed visceral hypersensitivity and clear signs of anhedonia in our SOY-exposed male rats, but no clear colonic motility alteration; therefore, these could be considered as a model of functional visceral pain. Some indications of immune dysfunctions were also evident, such as a tendency to higher numbers of mast cells in SOY-treated animals. We believe our preclinical results are potentially relevant in view of the high prevalence of functional gastrointestinal disorders and also the high prevalence of high-fat and calorie intake in occidental societies. Specific clinical assays are needed to clarify the connection between those two factors. We have addressed these issues in the Discussion and Conclusions, and hope it is now satisfactory.
[710] Are there any known differences in the modulation of abdominal pain between men and women by diet? Is ref. 82 a correct source in this matter?
ANSWER: As mentioned in the Discussion, there is a female “domination” in pain prevalence and perception. Women have lower pain thresholds for thermal, chemical, inflammatory and mechanical stimuli than males and more frequently report severe pain, more frequent bouts of pain, and more anatomically diffuse and longer lasting pain than men with the same disease type. However, mechanisms of pain modulation by diet considering gender and sex-related aspects seem to be unexplored. In our studies we documented that fat content and fatty acids composition affect colorectal sensitivity in male but not female rats which clearly suggest that there may be sexual dimorphism in pain perception regulated by diet. We hope these explanations are now satisfactory. Besides, reference 82 was removed, according to the reviewer’s suggestion.
[745] Here shall be more to elaborate on what type of dietary fiber and what daily dose is recommended as well as what the preferred fatty acid composition should be for IBS patients.
ANSWER: So far, our results suggest that HFD may induce diet-associated functional gastrointestinal disorders. Therefore, the next step will be to determine how this can be modified by other components also present in IBS, like stress, or to determine if particular functional foods (like those developed using coffee by-products, which is a project in process) may be helpful to alleviate/prevent the symptoms. Thus, although we do not have a definitive answer to these interesting questions at the moment, new specifically designed studies will be certainly useful in this context.
[835] Based on the presented research results, these conclusions remain unfounded.
ANSWER: As mentioned above, we have reviewed the whole text and particularly the conclusions. We believe the new conclusions are more adequate for our findings.
[838] Diet is certainly the main element in the prevention of FGID, however this is not entirely clear from the results presented in this article. In addition, the summary should include recommendations on which food ingredients are particularly important in relieving the symptoms of IBS and other pain-related diseases. Although the direction is right, most of the work is based on assumptions to the applicability on human.
ANSWER: Thank you very much for your comment. As mentioned above, our results suggest HFD may cause the development of different symptoms associated with the highly prevalent brain-gut axis disorders. We agree in that more applicability of our results is needed based on determining the exact components underlying our results, and also searching for other that might exert a relieving or preventative role. We believe our results provide a starting point for new more targeted research aimed at addressing these different issues. We have reviewed the conclusions to make them more directly related with our findings and to highlight the need to performe clinical studies to demonstrate their applicability.
More focus shall be paid to the development of the methodology of research during tests on rats in terms of effectiveness and reliability of expected results, before making conclusions relating to IBS diet correlations on human. Continuation of human clinical trials is imperative to confirm the findings presented in the study.
ANSWER: Thank you very much for your comment. We believe we have adequately addressed it in the previous answers and the new version of our manuscript. We hope it is now acceptable.
[866] (.); [868) (.); [895] (,/.)’ [909] (bold year); [1080] (bold year)
ANSWER: We have rephrased quite a bit part of the text and have tried to polish it in order to avoid mistakes like the previously found. We hope the text is now acceptable.
Round 2
Reviewer 1 Report
There is still need to be improved with this study. AIM 1 is considering the "fat proportion" between the SOY purified diet and the grain-based diet. While it is true that the purified diets are higher in energy density, due to the lower fiber content than the grain-based diets, the purified diets are not by any means high in fat. These purified diets are based on the AIN-93G diet (as the authors point out), which is typically used for maintaining a normal weight gain and provides for adequate growth of growing rodents after weaning and supports rodent health during pregnancy and lactation phases. While these purified diets are technically higher in fat compared to the grain based diet, this is a minimal difference, and actually they could say that all macronutrients are higher in concentration as both protein and digestible carbohydrate are higher. High fat diets are typically between 30 kcal% fat to 60 kcal% fat, which is in stark contrast to the AIN-93G diet which contains about 16 kcal% fat (or 7% by wt). Also, as pointed out in the previous review, there are so many differences between the grain-based diet and purified diet that it is not possible to conclude what nutrients are affecting the measures. To me, this is a fatal flaw in AIM 1 and it con not be stated that fat proportion is causing the effects.
The only diets that can really be compared are the 3 purified diets as they are matched with the only differences being the fat sources, so the AIM 2 is fine.
While the introduction was written better, it is still confusing in that the authors lead you to believe that they are going to uncover the pathophysiology of IBS, but they are not studying an IBS model, so it is misleading.
An argument was made by the authors that grain-based diets being more complex are in some way similar to a human diet. However, this diet doesn't allow for us to understand what the effect fat is having, and really, as pointed out above, there is a minimal difference in fat levels, so AIM 1 is really just comparing a purified low fat diet to a grain-based diet. The point of using a refined diet such as the AIN-93G in this study is to have a nutritionally defined diet that can be modified easily to understand how nutrient changes affect a given phenotype. This is why it was very suitable for studying AIM 2, and really why purified diets are important for preclinical studies as they are well defined and allow for easy revisions as each ingredient contains one main nutrient.
In the methods section Table 1B should provide more specific information on the fatty acids present in these oils. It would also be better if it provided a better idea of what is in each diet as they are including only 1/2 of the fat as coconut oil or evening primrose oil, so the proportion of the SFAs, MUFAs and PUFAs will be altered.
In line 180, they state that EP oil is high in SFAs, which is not true.
In line 775 and 776, this statement about fiber is not correct - grain-based diets typically have much more fiber (typically around 4 fold) than what is present in the AIN-93G (and AIN-93M) diets.
There is no need to describe different fatty acids in the introduction. This is a nutritional journal and readers will be aware of the differences between short, medium and long chain fatty acids and the differences between SFAs, MUFAs, and PUFAs.
Lines 785 - 786- the AIN-93G is not almost double the energy density of the grain-based diet.
Lines 856 and 857 should provide info that soybean oil contains around 50% linoleic acid and around 8% alpha-linolenic acid. This needs to be stated as they are mentioning the specifics for EP oil, which also contains those fatty acids.
Lines 881 - 886 has some far reaching statements. Again, there wasn't an IBD model being studied, so the relevance of the fatty acid composition on patients that are suffering from FGIDs, including IBS, is not possible to know with this study design.
Author Response
Dear reviewer,
Thank you very much for your comments and suggestions. We have tried to improve our manuscript following all your recommendations and hope it is now acceptable.
Please see attached our point-by-point answers.
Kind regards,
Raquel Abalo (on behalf of all co-authors)

Reviewer 2 Report
The authors have satisfactorily addressed all points of criticism. The paper has significantly gained quality and now represents an outstanding example of a well-designed, comprehensive, multi-faceted preclinical trial.
Author Response
Reviewer 2:
COMMENT 1: The authors have satisfactorily addressed all points of criticism. The paper has significantly gained quality and now represents an outstanding example of a well-designed, comprehensive, multi-faceted preclinical trial.
ANSWER: Thank you very much for your positive consideration towards our manuscript.